# Exploring Model Kinship for Merging Large Language Models

## Abstract

Model merging has become one of the key technologies for enhancing the capabilities and efficiency of Large Language Models (LLMs). However, our understanding of the expected performance gains and principles when merging any two models remains limited. In this work, we introduce *model kinship*, the degree of similarity or relatedness between LLMs, analogous to **biological evolution**. With comprehensive empirical analysis, we find that there is a certain relationship between model kinship and the performance gains after model merging, which can help guide our selection of candidate models. Inspired by this, we propose a new model merging strategy: Top-$k$ Greedy Merging with Model Kinship, which can yield better performance on benchmark datasets. Specifically, we discover that using model kinship as a criterion can assist us in continuously performing model merging, alleviating the degradation (local optima) in model evolution, whereas model kinship can serve as a guide to escape these traps.

## 1 Introduction

Fine-tuning pre-trained models (PTMs) for downstream tasks has become a popular practice, particularly demonstrating significant effectiveness in Large Language Models (LLMs) (Kolesnikov et al., 2020; Qiu et al., 2020; Askell et al., 2021; Ouyang et al., 2022; Zhao et al., 2023). However, deploying separate fine-tuned models for each task can be resource-intensive (Fifty et al., 2021), which drives the increasing demand for multitask learning solutions (Zhang & Yang, 2022; Lu et al., 2024; Liu et al., 2024). Recent studies suggest that model merging (Singh & Jaggi, 2020; Sung et al., 2023; Goddard et al., 2024; Matena & Raffel, 2022; Yang et al., 2024a) offers a viable approach for achieving multitask objectives by integrating multiple expert models. Furthermore, advancements in model merging toolkits (Goddard et al., 2024; Tang et al., 2024) enable users with limited expertise to easily conduct merging experiments, leading to an evolution of LLMs for the community.

To date, through model merging techniques, reseracheres have developed many more powerful LLMs through iterative model merging (Beeching et al., 2023), and to some extent, achieved model evolution (Figure 1(c)). Despite these successes, progress has predominantly relied on trial and error, along with extensive human expertise, but lacks formalized guidance and standardized procedures. As the merging iterations progress, achieving further generalization gains becomes increasingly challenging (More details in Section 3). For example, as shown in Figure 1, model merging often resembles **the process of hybrid evolution in biology**, where the next generation may not show significant improvements or may even regress, highlighting the imperative for a deeper exploration of the underlying mechanisms driving these advancements.

To address this, we introduce *model kinship*, a metric inspired by the concept of kinship (Sahlins, 2013) from evolutionary biology (Figure 1(a)). This metric is designed to estimate the degree of similarity or relatedness between LLMs during the iterative model merging process, offering insights intended to enhance the effectiveness of the merging strategy. We utilize the model kinship to conduct a comprehensive analysis of model merging experiments from two perspectives: the overall merging process, including various independent merge experiments and the evolution path of specific models, demonstrating the complete merging trajectory.

**Model kinship correlates with average performance gain in model merging.** Emperically, we find that there is a strong correlation between variations in multitask capability, estimated by average task performance, and model kinship, which can help guide our selection of candidate models.

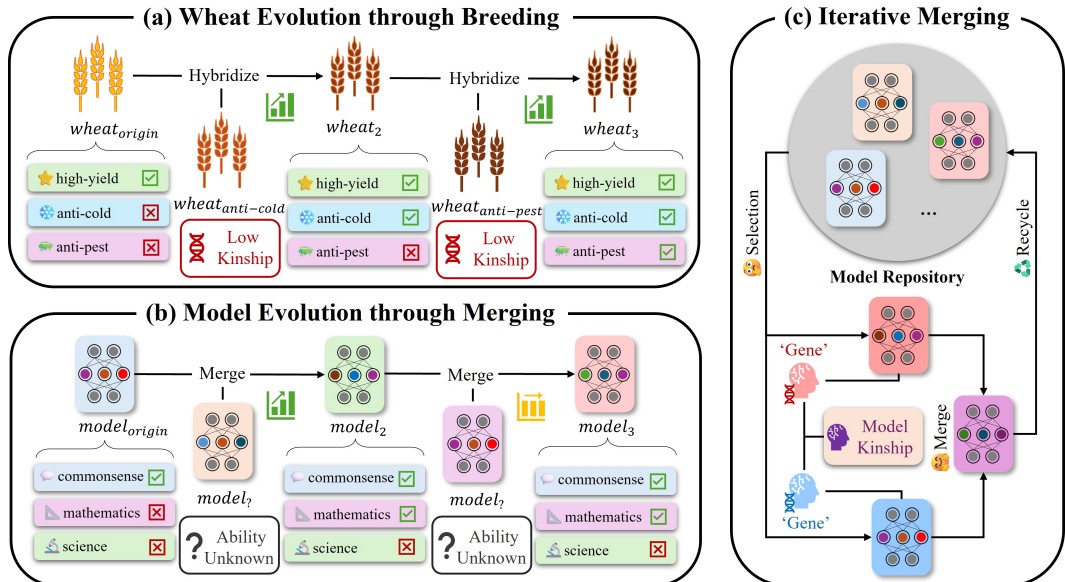

Figure 1: **An intuitive comparison between wheat evolution and model evolution**. An interesting parallel can be drawn between biological reproduction (**Part a**) and the process of model evolution (**Part b**). In biological systems, offspring inherit genetic material from both parents, forming a new genotype through the combination of parental traits. Similarly, in model merging, the merged model inherits parameters or weights from the contributing models. **Part c** demonstrates the iterative execution of model evolution. Starting with a group of LLMs, the repository evolves through a Selection-Merge-Recycle iteration. To be noted, *model kinship* can serve as an effective tool to guide this iterative model merging process (e.g., infer whether there may be gains after model merging.).

We also observe that the model merging process consists of two stages: the learning stage, where models experience significant performance improvements, and the saturation stage, where further improvements diminish and eventually stagnate. We think that the stagnation of improvements may be due to convergence in weight space, suggesting the presence of optimization challenges like local optima traps.

Inspired by this, we propose a new model merging strategy: Top-$k$ Greedy Merging with Model Kinship. Specifically, we find that leveraging model kinship as a criterion enables more effective model merging, helping to mitigate degradation and avoid local optima during model evolution. Model kinship also proves useful as an early stopping criterion, improving the efficiency of the merging process. Overall, this paper makes **three key contributions**:

1. **Introduction of *Model Kinship*:** We introduce model kinship, designed to assess the degree of similarity or relatedness between LLMs during the merging process, which can guide model merging strategies and holds promise for advancing auto-merging research.

2. **Empirical Analysis of Model Evolution:** We present a comprehensive empirical analysis of model evolution through iterative merging. Our findings highlight the dynamics of multitask performance improvement and stagnation. Additionally, we propose a preliminary explanation of the underlying mechanisms using model kinship.

3. **Practical Model Merging Strategies using *Model Kinship*:** We demostrate how model kinship guides the model merging process to tackle optimization challenges, and provide practical strategies: Top-$k$ Greedy Merging with Model Kinship, to enhance efficiency and effectiveness of model evolution.

## 2 BACKGROUND

### 2.1 MODEL MERGING: FUNDAMENTALS

Model merging aims to integrate two or more domain-specific models into a unified framework, thereby harnessing their compositive capabilities across multiple tasks (Sung et al., 2023). While this approach shares conceptual similarities with ensemble methods (Dietterich et al., 2002; Dong et al., 2020; Jiang et al., 2023b), model merging generates a single, generalized model, avoiding the increased inference time associated with ensembles. Let $f_i$ represent the $i$-th model for merging, each with its unique parameters $\theta_i$. If the merging process follows method $\mathcal{F}$, the prediction $\hat{y}$ of the merged model $f_{\text{merge}}$ for input $x$ is:

$$\hat{y} = f_{\text{merge}}(x) = \mathcal{F}(f_1(x; \theta_1), f_2(x; \theta_2), \ldots, f_n(x; \theta_n)) \tag{1}$$

### 2.2 ITERATIVE MERGING: EFFECTS AND CHALLENGES

Parameter averaging methods allow the merged model to retain the same architecture and parameter size as the original models, allowing for reuse in future merging processes. By benefiting from this feature, the community iteratively enhances models through repeated applications of model merging, a process we term **"Model Evolution"**. Empirical evidence from the open LLM leaderboard (Beeching et al., 2023) demonstrates that model evolution can produce highly generalized models, often surpassing those created through a single merging step (Maxime Labonne , 2024).

However, one of the main challenges limiting the effectiveness of iterative merging is the merging strategy. The community primarily relies on two approaches: **1) Task-Capability-Based Merging:** This approach uses task capabilities, as evaluated by benchmarking tools (Gao et al., 2024; Li et al., 2023c), to guide model evolution, compensating for one model's deficiencies by leveraging another's strengths. While effective in principle, this strategy heavily relies on human judgment and becomes impractical in complex merging scenarios involving more than two tasks. **2) Greedy Merging of Top-Performing Models:** This strategy involves merging the best-performing models with the expectation of producing an even better model. While widely applicable, it is inherently greedy and prone to getting stuck in local optima, as further discussed in Sections 3.4 and 4.2. Therefore, a problem raised.

> **Problem:** *Is there another strategy or metric we can use to better achieve model evolution?*

### 2.3 MODEL KINSHIP: CONCEPT AND FORMULATION

Considering the two strategies above, we are exploring a new approach that can identify task-related differences between models to maximize the outcomes of merging, without the need for costly evaluations. Drawing inspiration from the parallel between artificial selection and model evolution (as detailed in Appendix C), we hypothesize that a concept analogous to *kinship*, which is central to understanding breeding relationships in evolutionary biology (Thompson, 1985), can be applied. Therefore, we propose the concept of *model kinship*

Model Kinship builds upon the cosine similarity analysis introduced in Task Arithmetic paper (Ilharco et al., 2023). It is designed to evaluate the degree of similarity or relatedness between the task capabilities of large language models (LLMs) solely based on their "genetic" information (i.e., the changes in weights) during model evolution. Considering two models $m_i$, $m_j$ involved in a model evolution originated from the pre-trained model $m_{base}$, the weights of $m_i$, $m_j$ are denoted as $\theta_i, \theta_j \in \mathbb{R}^d$. Similarly, $\theta_{\text{base}} \in \mathbb{R}^d$ represents the weights of the pre-trained model. Since the differences between models emerge after fine-tuning and merging, the variation of weights during model evolution is crucial. It is calculated as:

$$\delta_i = \theta_i - \theta_{\text{base}}, \delta_j = \theta_j - \theta_{\text{base}} \tag{2}$$

Model kinship $r$ is designed to capture the similarity of task capabilities between models. In this paper, we explore multiple potential metrics for evaluating similarity. For the calculation, $sim(\cdot, \cdot)$ denotes the similarity metric function used. Considering two cases merging of 2 models and merging of $n$ models, we formally define model kinship $r$ as:

$$r = \begin{cases} sim(\delta_1, \delta_2), & \text{for merging 2 models} \\ \frac{2}{n(n-1)} \sum_{1 \leq i < j \leq n} sim(\delta_i, \delta_j), & \text{for merging } n \text{ models} \end{cases} \quad (3)$$

## 3 PRELIMINARY ANALYSIS OF MODEL KINSHIP

In this section, we present a preliminary analysis of community merging experiments on LLMs to explore how model kinship can inform and enhance model evolution.

### 3.1 EVALUATION METRICS

Let $T$ be the set of tasks in the task group, where $T = \{T_1, T_2, \ldots, T_n\}$. Each task $T_i$ in the set $T$ is associated with a performance measure $P_i$ for the LLM. For a multitask objective, the Average Task Performance (Avg.) $\bar{P}$ is calculated using the equation:

$$\bar{P} = \frac{1}{n} \sum_{i=1}^{n} P_i \quad (4)$$

To evaluate the effectiveness of a single merge, we propose the merge gain metric. Assume we have two models $m_{pre-1}$ and $m_{pre-2}$ and their average task performance are $\bar{P}_{pre-1}$ and $\bar{P}_{pre-2}$, intuitively, we believe the $\bar{P}_{\text{merged}}$ lie around the mean of $\bar{P}_{pre-1}$ and $\bar{P}_{pre-2}$. The merge gain is calculated as the difference of $\bar{P}_{\text{merged}}$ from the mean value of $\bar{P}_{pre-1}$ and $\bar{P}_{pre-2}$. For a merging recipe with $k$ models, the merge gain is:

$$Gain = \bar{P}_{\text{merged}} - \frac{1}{k} \sum_{i=1}^{k} \bar{P}_{\text{pre-i}} \quad (5)$$

In the following analysis, we use the task group T = {*ARC, HellaSwag, MMLU, TruthfulQA, Winogrande, GSM8K*}. All models are either fine-tuned or merged from the *Mistral-7B* architecture.

### 3.2 CORRELATION ANALYSIS OF MODEL KINSHIP AND PERFORMANCE GAIN

Table 1: **Correlation** of Model Kinship based on different correlation function $sim(\cdot, \cdot)$ with Merge Gain, along with their corresponding p-values.

| Metric | Correlation (Normal Value) | Correlation (Absolute Value) |
|---|---|---|
| PCC | -0.50 | **-0.59** |
| P-value | 0.063 | **0.023** |
| CS | -0.45 | **-0.66** |
| P-value | 0.098 | **0.008** |
| ED | 0.46 | **0.67** |
| P-value | 0.091 | **0.007** |

In this analysis, we examine the distribution of merge gain and model kinship based on *Pearson Correlation Coefficient (PCC), Cosine Similarity (CS)* and *Euclidean Distance (ED)* in open-sourced LLMs, originating from the *Mistral-7B* (Jiang et al., 2023a). Those models are obtained from the HuggingFace, with assistance from the Open LLM Leaderboard (Details in Appendix B.).

#### 3.2.1 RESULTS

Figure 2 illustrates the distribution of model kinship based on three similarity metrics (PCC, CS, ED) in relation to merge gain. The scatter plots reveal a moderate correlation between model kinship and merge gain, as indicated by the trend lines. To further quantify these relationships, the correlation value (use Pearson Correlation Coefficient) between model kinship and merge

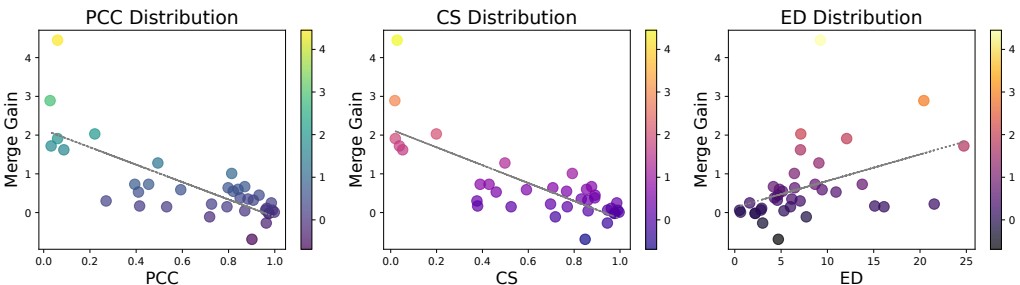

Figure 2: **Distribution of Sample Experiments**: Relationship Between Model Kinship (X-axis) and Merge Gain (Y-axis). Model Kinships are calculated using the Pearson Correlation Coefficient (PCC), Cosine Similarity (CS) and Euclidean Distance (ED).

gain are calculated, as detailed in the second column of Table 1. While moderate correlations are observed for all three metrics (negative correlation for PCC and CS, and positive correlation for ED), the corresponding p-values indicate a weak level of statistical significance, ranging from 0.05 to 0.1. In contrast, when examining absolute merge gain, we find stronger and statistically significant correlations, as shown in the third column of Table 1. These results suggest that model kinship alone is insufficient to predict whether a model can acquire enhanced generalized performance through merging. However, it may serve as a factor in determining the upper limit of merge gains, highlighting the potential outcomes of merging. Since no significant differences are observed among the three metrics, we will focus solely on model kinship based on PCC in the following sections to simplify the demonstration.

### 3.3 SEQUENCE ANALYSIS OF MODEL EVOLUTION PATHS

In this analysis, we examine changes in performance and model kinship across independent model evolution paths to identify the phased pattern of the merging process. We focus on the *yamshadow experiment 28-7B* (Labonne, 2024), a Mistral 7B architecture model ranked as the top 7B merged model on the Open LLM Leaderboard. From its model family tree, we extract two primary merging paths: **Path 1** and **Path 2**.

#### 3.3.1 RESULTS

We first focus on the average task performance and merge gains throughout the model evolution path (Figure 3.) Detailed data and branch information are summarized in Appendix B). Our observations indicate that the performance improvements of the iterative merging process are not linear and can be divided into two stages:

- **Learning Stage.** In this stage, the average task performance generally experiences a rapid increase. Noticeable merge gains suggest that the merged models are continually acquiring multitask capabilities through the merging process.

- **Saturation Stage.** As the process continues, improvements begin to plateau. During this stage, the merge gains approach zero, indicating that the model can no longer benefit from the merging process and has ceased to improve.

Additionally, we compare the trend of model kinship with average task performance. Figure 4 illustrates the changes in model kinship alongside average task performance (normalized to the same range as the corresponding metric) throughout the model evolution paths. We observe model kinship exhibits a similar stage-specific pattern, particularly evident in the saturation stage, suggesting a potential relationship with the underlying cause of saturation.

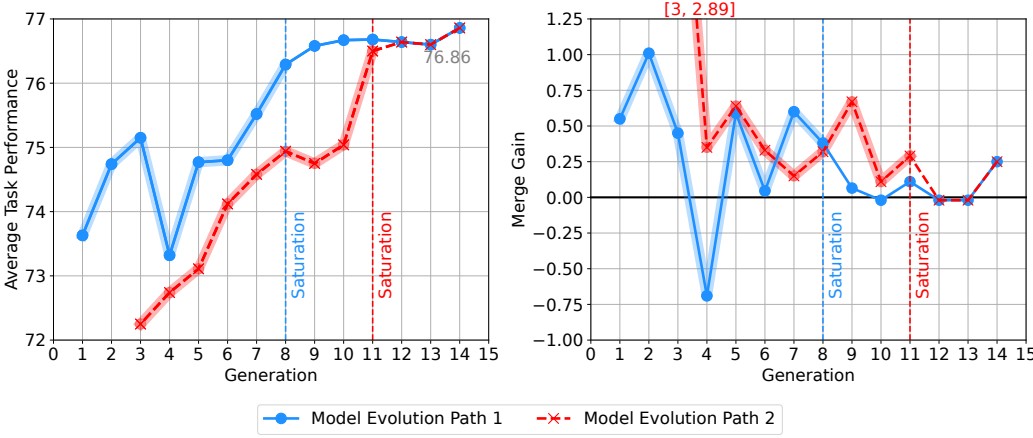

Figure 3: **Change in Average Task Performance and Merge Gain across the Model Evolution process:** The selected paths originate from two distinct initial models, with the saturation stage observed after the vertical line. Note that the generation of Path 2 is aligned with Path 1 for demonstration purposes.

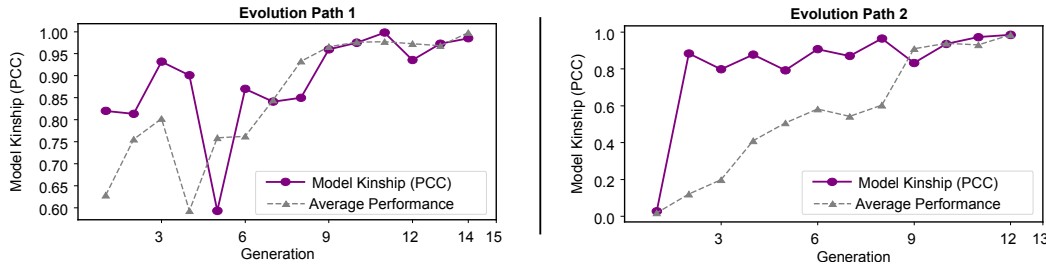

Figure 4: **Comparision** between Model Kinship (measured by Pearson Correlation Coefficient) and Average Task Performance (normalized to the same value scale).

### 3.4 ANALYSIS OF THE MODEL KINSHIP IN DIFFERENT MERGING STAGES

Findings in previous analysis reveals a initial observation in relationship between model kinship and model evolution. To further investigate the causality between model kinship and the stagnation of improvements, we examine the variation of model kinship across different merging stages from a broader perspective.

Given the community's predominant use of the performance-prior strategy, we calculate model kinship among models with similar performance, simulating the selection of top-performing models at each stage. For this analysis, we randomly select 5 models from each merging stage, as delineated by boundaries identified in prior analysis - Saturation Stage ($\geq 0.75$), Learning Stage ($<0.75$ and $\geq 0.73$), and Initial Merges (fine-tuned models) to form three foundation model groups, representing potential merges at different stages of model evolution.

Figure 5 illustrates the model kinship between models within each group. We observe that model kinship increases with the average task performance across models that follow different evolution paths. Additionally, during the saturation stage, all potential merges display a strong affinity, with model kinship values nearing 1. Since model kinship indicates the similarity of weights, we conclude the final findings as:

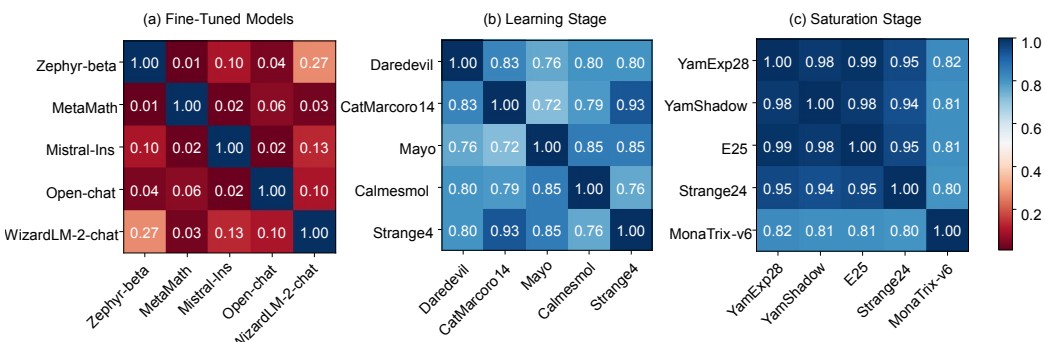

Figure 5: **The Model Kinship Matrices for the three model groups**. Each element represents the model kinship value between the corresponding models. In Group B and C, the merged models are arranged by average task performance, ordered from **high to low** (left to right).

> ***Findings:*** *Model merging experiences a saturation stage, where the model kinship among top-performing models increases throughout the iterative merging process. This implies that the models converge to similar forms, resulting in excessive relatedness that undermines the effectiveness of the model merging strategy.*

## 4 USING MODEL KINSHIP TO IMPROVE MODEL MERGING

Inspired by the above findings, we further leverage model kinship to enhance the model merging process. We firstly conduct experiments employing a performance-prior greedy merging strategy. Note that the greedy strategy may eventually lead to convergence. To address this, we further introduce Top-$k$ Greedy Merging with Model Kinship (Algorithm 1). Our results indicate that while the greedy strategy focuses on short-term gains, it can lead to parameter convergence and suboptimal outcomes. By integrating model kinship, we can help the strategy avoid local optima. Furthermore, we find that model kinship holds potential for enhancing merging strategies as an early stopping criterion.

### 4.1 EXPERIMENT SETUP

**LLMs.** We select three fine-tuned, open-source LLMs based on the *Mistral-7B* architecture from HuggingFace: ***mistral-7b-instruct-v0.2***, ***metamath-mistral-7b***, and ***open-chat-3.5-1210***.

**Datasets.** Evaluation is conducted using three task-specific benchmark datasets: Winogrande, GSM8k, and TruthfulQA.[1] These benchmarks demonstrate the distinct strengths of the three selected fine-tuned models. Further details on the tasks are provided in Appendix B.4.

**Merging Method.** We conduct two iterative model merging experiments, both utilizing the SLERP (Spherical Linear Interpolation) (Shoemake, 1985) for the single merging step. For implementation, we employ Mergekit (Goddard et al., 2024), a comprehensive toolkit that offers simple access to state-of-the-art model merging techniques.

**Top $k$ Greedy Merging.** This strategy utilizes the vanilla Top-$k$ Greedy Merging approach on $n$ LLMs (as outlined in the black section of Algorithm 1). This approach has been widely adopted in the community and has demonstrated notable success. In Figure 6 (b), models generated by the greedy strategy are indicated in green, while the best-performing models are highlighted in red.

---

[1]The evaluation configurations are as follows: Winogrande (5-shot), GSM8K (5-shot), and TruthfulQA MC2 (0-shot). We utilize the Language Model Evaluation Harness (Gao et al., 2024), a widely adopted framework for testing LLMs.

---

**Algorithm 1** Top $k$ Greedy Merging with Model Kinship.

---

**Require:** A set $M$ of $n$ foundation models $\{m_1, m_2, \ldots, m_n\}$, Evaluation function $f$, Similarity metric function $sim(\cdot, \cdot)$ for model kinship.

1: Generate the first generation of merged models $G_1$ by merging each pair in set $M$, and set gneration $i = 1$.
2: Combine the set $G_1$ into set $M$.
3: Evaluate each model $m$ in set $M$, and select the top $k$ models. Denote this set as $S = \{m_1, m_2, \ldots, m_k\}$.
4: Initialize a variable $S_{\text{prev}} = \emptyset$ to store the top $k$ models from the previous iteration.
5: **while** $S \neq S_{\text{prev}}$ **do**
6:    i++
7:    Set $S_{\text{prev}} = S$.
8:    Select each model pair from $S$. Denote this set as $P = \{p_1, p_2, \ldots, p_j\}$.
9:    Merge every selected pair in set $P$ as merged model set $G_i = \{m_1, m_2, \ldots, m_j\}$ for generation $i$, and add each merged model into set $M$.
10:    Identify the current best model $m_{best} \in S$.
11:    Identify the model $m_f \in S$ with the lowest model kinship to $m_{best}$ from the $G_{i-1}$ according to the similarity metric $sim(\cdot, \cdot)$.
12:    Merge $m_f$ with $m_{best}$ to generate a new model $m_{\text{exp}}$, and add $m_{\text{exp}}$ into set $G_i$ and set $M$.
13:    Evaluate each new model $m \in G_i$ using $f$ and update $S$.
14:    Evaluate $m_{\text{exp}}$ using $f$ and update $S$.
15: **end while**

---

**Note:** The blue-highlighted steps are only executed in modified experiments incorporating model kinship-based exploration. To distinguish between different models in the subsequent experiments, each model generated in a given generation is named as **model-generation-id**.

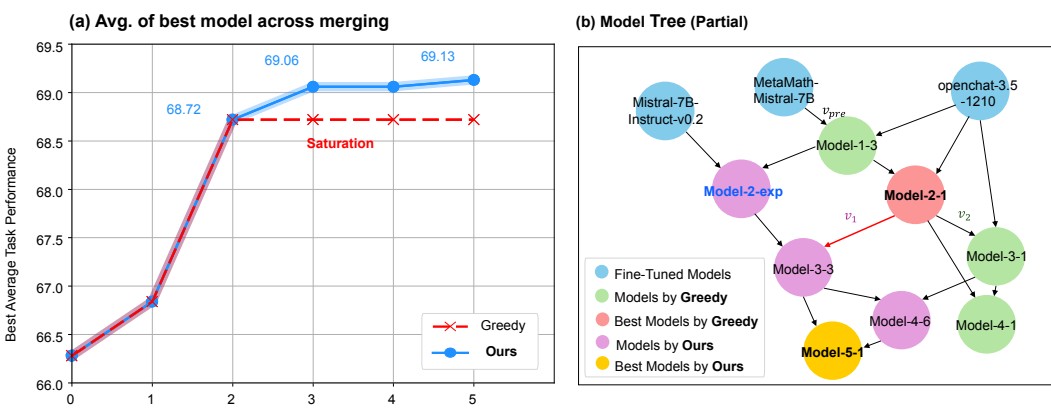

Figure 6: **Left (a)**: The comparison of task performance improvement across merging generations. The **red curve** (greedy strategy) saturates by generation 2, while the **blue curve** (modified strategy) escapes the local optima at generation 2 and continues improving multitask capabilities. **Right (b)**: The partial model family tree from the controled experiments. The **red arrow** shows the critical change between experiment 1 and experiment 2 in the evolution path.

**Top $k$ Greedy Merging with Model Kinship.** The propposed strategy simply introduces an additional exploration step, based on model kinship, to the original greedy strategy (highlighted by the blue part in Algorithm 1). This approach aims to merge the best-performing model with the model that has the most distinct task capabilities, in order to discover potentially better solutions. In Figure 6 (b), models generated by our strategy are marked in purple, while the best-performing models are marked in yellow.

## 4.2 RESULTS AND DISCUSSION

Table 2: Results of merging experiments comparing the vanilla greedy strategy and our proposed approach. The first three models serve as the foundation models in both experiments.**Note**: The model kinship experiment was terminated at generation 5, as it has already outperformed the greedy strategy by that point.

| Greedy Strategy | | | | + Model Kinship | | | |
|---|---|---|---|---|---|---|---|
| **Model** | **Avg.** | **Gain** | **Kinship** | **Model** | **Avg.** | **Gain** | **Kinship** |
| **MetaMath** | 63.72 | / | / | **MetaMath** | 63.72 | / | / |
| **Instruct** | 61.82 | / | / | **Instruct** | 61.82 | / | / |
| **Open-chat** | 66.28 | / | / | **Open-chat** | 66.28 | / | / |
| model-1-1 | 62.17 | -0.6 | 0.01 | model-1-1 | 62.17 | -0.6 | 0.01 |
| model-1-2 | 64.02 | -0.03 | -0.02 | model-1-2 | 64.02 | -0.03 | -0.02 |
| model-1-3 | 66.84 | +1.84 | 0.05 | model-1-3 | 66.84 | +1.84 | 0.05 |
| **model-2-1** | **68.72** | **+2.16** | **0.93** | model-2-1 | 68.72 | +2.16 | 0.93 |
| model-2-2 | 61.47 | -3.96 | 0.57 | model-2-2 | 61.47 | -3.96 | 0.57 |
| model-2-3 | 61.32 | -3.83 | 0.58 | model-2-3 | 61.32 | -3.83 | 0.58 |
| model-3-1 | 68.59 | +1.09 | 0.95 | model-3-2 | 67.74 | +1.09 | 0.93 |
| model-3-2 | 67.74 | -0.04 | 0.93 | model-3-3 | 69.06 | +0.74 | 0.24 |
| - | - | - | - | model-3-4 | 68.61 | +1.13 | 0.32 |
| model-4-1 | 68.51 | -0.14 | 0.98 | model-4-4 | 68.75 | -0.14 | 0.54 |
| model-4-2 | 68.04 | -0.19 | 0.98 | model-4-5 | 68.39 | -0.27 | 0.66 |
| model-4-3 | 68.53 | +0.37 | 0.94 | model-4-6 | 69.03 | +0.15 | 0.52 |
| | - | - | - | **model-5-1** | **69.13** | **+0.04** | **0.65** |
| | - | - | - | model-5-2 | 68.98 | +0.07 | 0.65 |
| | - | - | - | model-5-3 | 68.63 | -0.37 | 0.98 |

Figure 6 (a) illustrates the improvements in top average task performance across merging generations. Table 2 provides the model average task performance, merge gain, and model kinship for each generation, comparing the original greedy merging strategy with our kinship-based method. Both strategies achieve the multitask goals. However, the *vanilla greedy strategy* stops improving after Generation 2, stabilizing at an average task performance of **68.72**. In contrast, Experiment 2, utilizing model kinship-based exploration, escapes the local optima (Model-2-1) and continues to improve, reaching **69.13** by Generation 5.

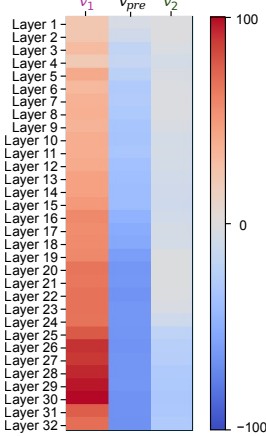

Figure 7: **Weight Change**.

**Merging Models with Low Kinship can Boost Exploration.** Figure 6 (b) highlights the key branch of the model family tree. To investigate how merging models with low kinship helps escape local optima, we focus on the bifurcation point and analyze the weight changes: $v_1$ (from *Model-2-1* to *Model-3-1*) and $v_2$ (from *Model-2-1* to *Model-3-3*) in two separate experiments. The previous weight change, $v_{pre}$ (from *Model-1-3* to *Model-2-1*), serves as a baseline. Figure 7 reveals that merging with the exploration model resulted in significant weight changes in a distinct direction, introducing novel variations into the weight space. In contrast, $v_1$ shows minimal weight change, as the merging effect is reduced due to the high similarity between the weights of *openchat-3.5* and *Model-2-1*.

**Early Stopping at High Kinship can Improve Efficiency.** We observe that the saturation stage of model evolution is particularly resource-intensive. In community experiments, **5 out of 14** merges in evolution path 1 resulted in an average improvement of just **0.57**, while **3 out of 12** merges in evolution Path 2 yields an average improvement of **0.36**. In our own experiments, applying a greedy strategy to a simple task lead to saturation after **2 out of 4** merges, with no further gains. These results indicate that human judgment and conventional stopping conditions cannot effectively halt

the merging process at the optimal time. Therefore, we propose that model kinship can be used as an effective early stopping signal. When merging converges, the model kinship between top-performing models often exceeds **0.9**. By halting the merging process at this point, time efficiency improves by approximately **30%**, with minimal or no reduction in performance.

## 5 RELATED WORK

Weight averaging is one of the most widely used techniques in model merging, with its origins traced back to Utans (1996), who first applied it in neural networks to achieve performance comparable to ensemble methods. Since the 2010s, weight averaging has found numerous applications in deep neural networks, including combining checkpoints to enhance the training process (Nagarajan & Kolter, 2019; Tarvainen & Valpola, 2017; Izmailov et al., 2018; Li et al., 2023b; Stoica et al., 2023; Padmanabhan et al., 2023; Jang et al., 2023), leveraging task-specific information (Li et al., 2023a; Smith & Gashler, 2017; Ilharco et al., 2022; Izmailov et al., 2018), and parallel training of large language models (LLMs) (Li et al., 2022). Discovery of Linear Mode Connectivity (LMC) (Garipov et al., 2018; Frankle et al., 2020; Entezari et al., 2022) further expands the use of weight averaging in fusing fine-tuned models through averaging methods (Neyshabur et al., 2020; Wortsman et al., 2022). Further studies have explored optimizable weights for merging, such as Fisher-Merging (Matena & Raffel, 2022), RegMean (Jin et al., 2023), AdaMerging (Yang et al., 2024b), MaTS (Tam et al., 2024). Ilharco et al. (2023) introduce task vectors, representing the weight difference between a fine-tuned model and its base. They demonstrate that arithmetic operations on these vectors enable model editing, such as achieving multitask learning. Further research on parameter interference led to TIES (Yadav et al., 2023), which preserves important weights and reduces sign conflicts, and DARE (Yu et al., 2024), which prevents interference by randomly dropping weights. The Model Breadcrumbs (Davari & Belilovsky, 2023) show that the removal of outliers in parameters can reduce noise in model merging. Merging models with different initializations requires additional considerations. Common methods exploit the permutation symmetry of neural networks (Ainsworth et al., 2022; Tatro et al., 2020; Singh & Jaggi, 2020; Guerrero-Peña et al., 2023), using alignment techniques to mitigate the interpolation barrier (Xu et al., 2024; Navon et al., 2024). While weight averaging cannot be directly applied to models with different architectures, it can still be used to enhance feasible fusion methods. Recent work, such as FuseChat (Wan et al., 2024b), combines weight averaging with Knowledge Fusion (Wan et al., 2024a) to develop innovative fusion techniques.

Recently, there have been some works exploring "model evolution". Tellamekala et al. (2024) propose the CoLD Fusion method, showing that iterative fusion can create effective multitask models. Labonne (2024) develop a tool to automatically merge models on Hugging Face, using an "Automerge" experiment to explore metrics in the merging process. Akiba et al. (2024) introduce Evolutionary Model Merge, employing evolutionary techniques to optimize model combinations, arguing that human intuition alone cannot uncover hidden patterns in merging.

## 6 CONCLUSION AND LIMITATIONS

In this paper, we introduce model kinship, the degree of similarity or relatedness between LLMs, for merging LLMs, which can help guide our selection of candidate models. We conduct comprehensive experiments to demonstrate its effectiveness in understanding the model evolution process. We further propose a new model merging strategy: Top-$k$ Greedy Merging with Model Kinship. We show that model kinship plays a crucial role in model evolution by guiding the process to escape local optima traps (in saturation stage), enabling further improvements. Additionally, we demonstrate that model kinship can detect the onset of convergence, allowing for early stopping and reducing the waste of computational resources in the merging process.

In a broad sense, our work explores how models can achieve autonomous evolution through model merging. Model merging can, to some extent, be likened to biological hybridization. Biological organisms have undergone billions of years of evolution to reach their current state. However, how silicon-based intelligence, represented by LLMs, evolves remains an unresolved mystery. We aspire that this work offer guidance and insights for the future merging and evolution of LLMs.

## REPRODUCIBILITY STATEMENT

The experimental setup can be found in Section 4.1. All model checkpoints are available on Hugging-Face, with detailed information provided in Appendices B.

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

## A    LIMITATIONS

However, there are several limitations to consider: *a)* The experiments in this study are conducted on models with the two architecture, leaving uncertainty about the transferability of our metric and method to other architectures, such as *Mamba* (Gu & Dao, 2023). *b)* The analysis relies on open-source data from the Open Leaderboard, which is community-generated and may contain noise due to user bias. *c)* Correlation metrics for model kinship have not been fully explored. Other metrics may perform better than those discussed in this paper. *d)* The effectiveness of model kinship is demonstrated through empirical evidence. However, a theoretical framework (such as the assumptions in Appendix C) is needed to explain model evolution and model kinship more rigorously. *e)* Model kinship currently guides merging and improves performance limits but does not support sustained evolution. Future progress may require environmental feedback, reward models (Silver et al., 2021), as well as new architectures.

## B    DETAILS OF EXPERIMENTS

All merged models from these experiments are accessible through the Hugging Face Hub[2]. The following tables cover two primary aspects:

- **(1)** Information on the selected model family trees for two distinct evolution paths, along with detailed analysis results for each merge.
- **(2)** A summary of the merge experiments conducted for distribution analysis.

### B.1    SELECTING THE EVOLUTION PATH

The evolution paths are selected using a structured process, focusing on identifying key sequences within the model family trees. The steps were as follows:

- **Model Family Tree Construction**: The complete model family tree is constructed by referencing model card details for each model involved.

---

[2]https://huggingface.co/datasets

- **Branch Identification**: We identified the two longest branches within each tree, representing significant sequences of model merging.

- **Performance and Kinship Evaluation**: These branches were analyzed for changes in merging performance, particularly focusing on shifts in multitask capabilities and model kinship metrics.

Table 3 and 4 present detailed information on the sequential merging process. The second and third columns record the foundational models involved in each merge, while the final column lists the resulting merged models.

Table 3: Model Family tree of evolution Path 1.

| Gen | Model-1 | Model-2 | Model-Merged |
|---|---|---|---|
| 1 | Marcoroni-7B-v3 | Mistral-7B-Merge-14-v0.1 | distilabeled-Marcoro14-7B-slerp |
| 2 | distilabeled-Marcoro14-7B | UNA-TheBeagle-7b-v1 | Beagle14-7B |
| 3 | NeuralBeagle14-7B | Turdus | TurdusBeagle-7B |
| 4 | TurdusBeagle-7B | FernandoGPT-v1 | StrangeMerges_9-7B-dare_ties |
| 5 | StrangeMerges_9-7B-dare_ties | MBX-7B-v3 | StrangeMerges_10-7B-slerp |
| 6 | StrangeMerges_10-7B-slerp | NeuralBeagle14-7B | StrangeMerges_11-7B-slerp |
| 7 | StrangeMerges_11-7B-slerp | MBX-7B-v3 | StrangeMerges_20-7B-slerp |
| 8 | StrangeMerges_20-7B-slerp | NeuTrixOmniBe-7B-model | StrangeMerges_21-7B-slerp |
| 9 | StrangeMerges_21-7B-slerp | Experiment26 | StrangeMerges_30-7B-slerp |
| 10 | StrangeMerges_30-7B-slerp | Experiment24 | StrangeMerges_31-7B-slerp |
| 11 | StrangeMerges_31-7B-slerp | Experiment28 | StrangeMerges_32-7B-slerp |
| 12 | StrangeMerges_32-7B-slerp | ... | shadow-clown-7B-slerp |
| 13 | shadow-clown-7B-slerp | yam-jom-7B | YamShadow-7B |
| 14 | YamShadow-7B | Experiment28 | YamshadowExperiment28-7B |

Table 4: Model Family tree of evolution Path 2.

| Gen | Model-1 | Model-2 | Model-Merged |
|---|---|---|---|
| 1 | neural-chat-7b-v3-3 | openchat-3.5-1210 | CatPPT-base |
| 2 | Marcoroni-7B-v3 | CatPPT-base | CatMacaroni-Slerp |
| 3 | LeoScorpius-7B | CatMacaroni-Slerp | SamirGPT-v1 |
| 4 | SamirGPT-v1 | ... | Daredevil-7B |
| 5 | NeuralBeagle14-7B | NeuralDaredevil-7B | DareBeagle-7B |
| 6 | Turdus | DareBeagle-7B | TurdusDareBeagle-7B |
| 7 | MarcMistral-7B | TurdusDareBeagle-7B | MarcDareBeagle-7B |
| 8 | MarcBeagle-7B | MarcDareBeagle-7B | MBX-7B |
| 9 | MBX-7B | ... | pastiche-crown-clown-7b-dare |
| 10 | pastiche-crown-clown-7b-dare | ... | shadow-clown-7B-slerp |
| 11 | yam-jom-7B | shadow-clown-7B-slerp | YamShadow-7B |
| 12 | Experiment28-7B | YamShadow-7B | YamshadowExperiment28-7B |

## B.2 ADDITIONAL RESULTS IN ANALYSIS

Table 5 and Table 6 present detailed analysis results that are not reported in the main paper. These include Average Task Performance (ATP), merge gains, and model kinship values, which are computed using Pearson Correlation coefficient, Cosine Similarity, and Euclidean Distance for each merge.

Table 5: Summary of Path 1 Results.

| Gen | Model-Merged | ATP | Gain | PCC | CS | ED |
|---|---|---|---|---|---|---|
| 1 | distilabeled-Marcoro14-7B-slerp | 73.63 | 0.55 | 0.82 | 0.76 | 5.15 |
| 2 | Beagle14-7B | 74.74 | 1.01 | 0.81 | 0.79 | 6.43 |
| 3 | StrangeMerges_9-7B-dare_ties | 75.15 | 0.45 | 0.93 | 0.89 | 4.66 |
| 4 | StrangeMerges_9-7B-dare_ties | 73.32 | -0.69 | 0.90 | 0.84 | 4.70 |
| 5 | StrangeMerges_10-7B-slerp | 74.77 | 0.59 | 0.59 | 0.59 | 9.43 |
| 6 | StrangeMerges_11-7B-slerp | 74.8 | 0.045 | 0.87 | 0.86 | 5.31 |
| 7 | StrangeMerges_20-7B-slerp | 75.52 | 0.6 | 0.84 | 0.85 | 4.82 |
| 8 | StrangeMerges_21-7B-slerp | 76.29 | 0.38 | 0.85 | 0.89 | 4.28 |
| 9 | StrangeMerges_30-7B-slerp | 76.58 | 0.065 | 0.96 | 0.94 | 2.83 |
| 10 | StrangeMerges_31-7B-slerp | 76.67 | -0.02 | 0.97 | 0.97 | 2.21 |
| 11 | StrangeMerges_32-7B-slerp | 76.68 | 0.11 | 0.99 | 0.99 | 0.62 |
| 12 | shadow-clown-7B-slerp | 76.64 | -0.02 | 0.93 | 0.94 | 2.49 |
| 13 | YamShadow-7B | 76.6 | -0.02 | 0.97 | 0.97 | 2.19 |
| 14 | YamshadowExperiment28-7B | 76.86 | 0.25 | 0.98 | 0.98 | 1.61 |

Table 6: Summary of Path 2 Results.

| Gen | Model-Merged | ATP | Gain | PCC | CS | ED |
|---|---|---|---|---|---|---|
| 1 | CatPPT-base | 72.25 | 2.89 | 0.02 | 0.01 | 20.41 |
| 2 | CatMacaroni-Slerp | 72.74 | 0.35 | 0.88 | 0.83 | 6.16 |
| 3 | SamirGPT-v1 | 73.11 | 0.64 | 0.79 | 0.70 | 6.47 |
| 4 | Daredevil-7B | 74.12 | 0.33 | 0.87 | 0.83 | 4.81 |
| 5 | DareBeagle-7B | 74.58 | 0.15 | 0.79 | 0.77 | 6.01 |
| 6 | TurdusDareBeagle-7B | 74.94 | 0.32 | 0.90 | 0.86 | 4.59 |
| 7 | MarcDareBeagle-7B | 74.75 | 0.67 | 0.87 | 0.87 | 4.17 |
| 8 | MBX-7B | 75.04 | 0.11 | 0.96 | 0.96 | 2.90 |
| 9 | pastiche-crown-clown-7b-dare | 76.50 | 0.29 | 0.83 | 0.84 | 5.38 |
| 10 | shadow-clown-7B-slerp | 76.64 | -0.02 | 0.93 | 0.94 | 2.49 |
| 11 | YamShadow-7B | 76.60 | -0.02 | 0.97 | 0.97 | 2.19 |
| 12 | YamshadowExperiment28-7B | 76.86 | 0.25 | 0.98 | 0.98 | 1.61 |

Table 7 presents all merge experiments contributing to the distribution analysis. The selection of sample experiments adheres to two rules: **(1)** Samples are evenly chosen across average task performance values ranging from *0.7* to *0.7686* (the average task performance of the best 7B merged model) to accurately reflect the full scope of model evolution. **(2)** The experiments involve merges of two foundation models, as including multiple models introduces excessive noise.

## B.3 DETAILS OF MODEL GROUP SELECTION

This appendix presents the exact models included in each model group, as shown in Table 8. The selection process is conducted across three distinct groups: **(1)** the top 5 models on the leaderboard, with a performance difference of 0.2, **(2)** 5 models with performance scores around 73 and a performance difference of 0.2, and **(3)** 5 fine-tuned models. It is important to note that the fine-tuned models were not selected based on performance, and may exhibit significant differences in results.

Table 7: **All Sample Experiments** used in distribution analysis.

| Model 1 | Model 2 | Merge Gain |
|---|---|---|
| Multi_verse_model-7B | Experiment26-7B | 0.06 |
| M7-7b | StrangeMerges_32-7B-slerp | -0.03 |
| Ognoexperiment27 | Multi_verse_model-7B | 0.03 |
| YamShadow-7B | Experiment28 | 0.25 |
| shadow-clown-7B-slerp | yam-jom-7B | -0.02 |
| StrangeMerges_21-7B-slerp | Experiment26 | 0.06 |
| StrangeMerges_31-7B-slerp | Experiment28 | 0.11 |
| NeuralBeagle14-7B | Turdus | 0.45 |
| DareBeagle-7B | Turdus | 0.32 |
| TurdusBeagle-7B | FernandoGPT-v1 | -0.69 |
| StrangeMerges_10-7B-slerp | NeuralBeagle14-7B | 0.04 |
| TurdusDareBeagle-7B | MarcMistral-7B | 0.67 |
| StrangeMerges_20-7B-slerp | NeuTrixOmniBe-7B-model-remix | 0.38 |
| StrangeMerges_11-7B-slerp | MBX-7B-v3 | 0.6 |
| Marcoroni-7B-v3 | Mistral-7B-Merge-14-v0.1 | 0.55 |
| distilabeled-Marcoro14-7B-slerp | UNA-TheBeagle-7b-v1 | 1.01 |
| UNA-TheBeagle-7b-v1 | distilabeled-Marcoro14-7B-slerp | 0.89 |
| CatPPT-base | Marcoroni-7B-v3 | 0.35 |
| CatMacaroni-Slerp | LeoScorpius-7B | 0.64 |
| NeuralDaredevil-7B | NeuralBeagle14-7B | 0.15 |
| StrangeMerges_9-7B-dare_ties | MBX-7B-v3 | 0.59 |
| mistral-ft-optimized-1218 | NeuralHerems-Mistral-2.5-7B | -0.85 |
| neural-chat-7b-v3-2 | OpenHermes-2.5-Mistral-7B | 1.91 |
| StrangeMerges_30-7B-slerp | Experiment24 | -0.02 |
| openchat-3.5-1210 | neural-chat-7b-v3-3 | 2.89 |
| MultiverseEx26-7B-slerp | CalmExperiment-7B-slerp | -0.09 |
| CapybaraMarcoroni-7B | DistilHermes-2.5-Mistral-7B | 0.47 |
| Multi_verse_model-7B | Calme-7B-Instruct-v0.9 | 0.04 |
| StrangeMerges_16-7B-slerp | coven_7b_128k_orpo_alpha | -0.35 |
| Kunoichi-DPO-v2-7B | AlphaMonarch-7B | -1.05 |
| StrangeMerges_15-7B-slerp | Kunoichi-7B | 0.39 |
| Mistral-T5-7B-v1 | Marcoroni-neural-chat-7B-v2 | -0.18 |
| Marcoro14-7B-slerp | mistral-ft-optimized-1218 | -0.61 |
| mistral-ft-optimized-1218 | NeuralHermes-2.5-Mistral-7B | -0.85 |
| MarcDareBeagle-7B | MarcBeagle-7B | -0.07 |
| MetaMath-Mistral-7B | Tulpar-7b-v2 | -0.29 |
| YugoGPT | AlphaMonarch-7B | -5.96 |

### B.4 DETAILS OF DATASETS SELECTION

In the main experiments, we utilize three task-specific benchmark datasets—Winogrande, GSM8k, and TruthfulQA—to evaluate the distinct strengths of the models. These datasets assess the following capabilities:

- **Winogrande**: Evaluates the model's commonsense reasoning.
- **GSM8k**: Measures the model's mathematical reasoning.
- **TruthfulQA**: Assesses the model's ability to identify and address human falsehoods.

## C ASSUMPTION OF CONTINUAL MODEL MERGING

Our findings in the main paper offer a new perspective on model evolution through multiple merging. If the merging process can be improved using a common optimization strategy, it raises the question of *whether the underlying mechanism mirrors this optimization problem.* Thus, we hypothesize the following:

Table 8: Model Group in Kinship Matrix Analysis.

| Group | Models |
|---|---|
| Top Model Group | YamshadowExperiment28-7B
Yamshadow-7B
Experiment25-7B
StrangeMerges_24-7B-slerp
MonaTrix-v6 |
| Mid Stage Model Group | Daredevil-7B
CatMarcoro14-7B
Mayo
Calmesmol-7B-slerp
StrangeMerges_4-7B-slerp |
| Fine-tuned Model Group | Zephyr-beta
MetaMath-Mistral-7B
Mistral-7B-Instruct-v0.2
openchat-3.5-1210
WizardLM-2 |

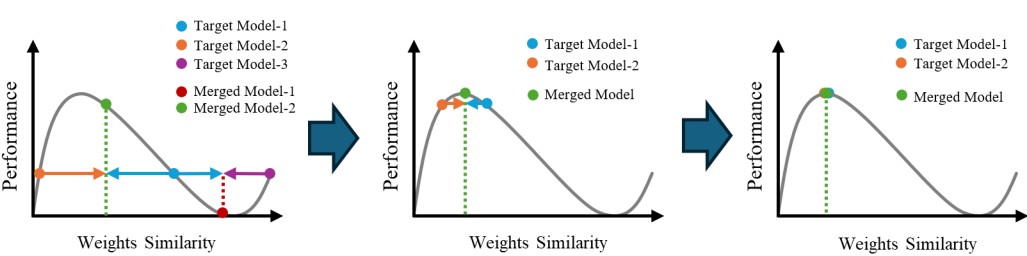

Figure 8: An intuitive illustration of **the optimization process assumption** in model evolution, where models progressively converge towards the optimal model.

> **Hypothesis:** *The evolution process may be simplified to a binary search process for most weight-averaging-based model merging methods.*

Figure 8 illustrates the ideal scenario in our assumption where multiple merges produce a highly generalized model. For the generalization task $t$, the y-axis represents the model performance for task $t$ and the x-axis represents the model's weight space. In early merging stages, models fine-tuned with different tasks exhibit significant weight space dissimilarity. The merging process averages these weight spaces, and the experiment conductor selects the better-merged models while discarding the inferior ones. In stage 2, the search area narrows and the improvements become stable, eventually leading to an optimized state in stage 3 when "saturation stage" occurs.

In this context, Model Kinship serves as a metric to quantify the weight space distance between two models, with a higher model kinship indicating a lower weight space distance. Following this assumption, our findings of the optimization problem in model evolution can be elucidated in Figure 9.

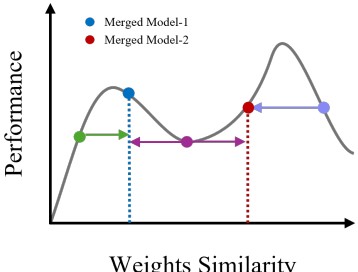

Figure 9: An intuitive illustration of **how model evolution can fall into local optima** due to a performance-prior strategy. It shows that Merged Model 2 may be overlooked, despite its potential for better multitask performance.

# D ADDITIONAL RESULTS: ANALYSIS OF MODEL KINSHIP AND AVERAGE TASK PERFORMANCE

This section provides supplementary analysis on the relationship between model kinship and average task performance. Figure 10 illustrates a comparison between average task performance and model kinship using two additional metrics not included in the main paper. From an intuitive observation, model kinship based on the three metrics exhibits a similar correlation with average task performance.

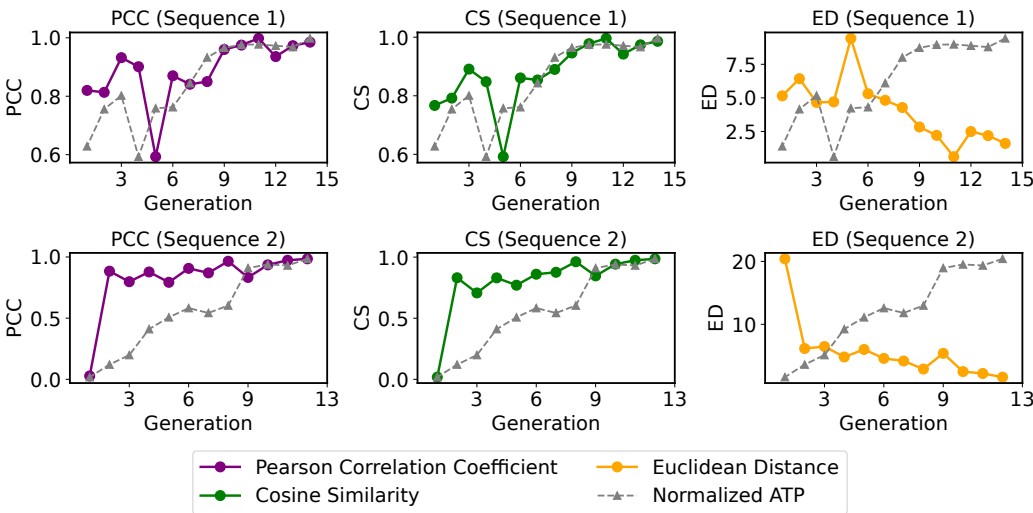

Figure 10: Illustration of comparison between the correlation of Pearson Correlation Coefficient (PCC), Cosine Similarity (CS), and Euclidean Distance (ED) with average task performance (Normalized to the same value scale).

# E REFERENCED CONCEPTS IN EVOLUTIONARY BIOLOGY

In this section, we detail the conceptual parallels between biological processes and model merging, highlighting our motivation for employing model kinship.

## E.1 ITERATIVE MERGING VS. ARTIFICIAL SELECTION

We draw inspiration for model evolution from biological evolution, specifically focusing on the correlation between biological evolution through artificial selection and model evolution via model merging. Artificial selection involves retaining desirable traits by manually selecting breeding pairs in each generation, typically those exhibiting the most significant features. Similarly, model evolution, as explored in this paper through Iterative Model Merging, adopts a comparable approach: users preserve desired task capabilities by strategically selecting merging pairs. Through iterative merging, they can develop a model proficient across all tasks in a given task set. To illustrate this comparison more effectively, Figure 11 depicts example of combining two features/task capabilities in evolution.

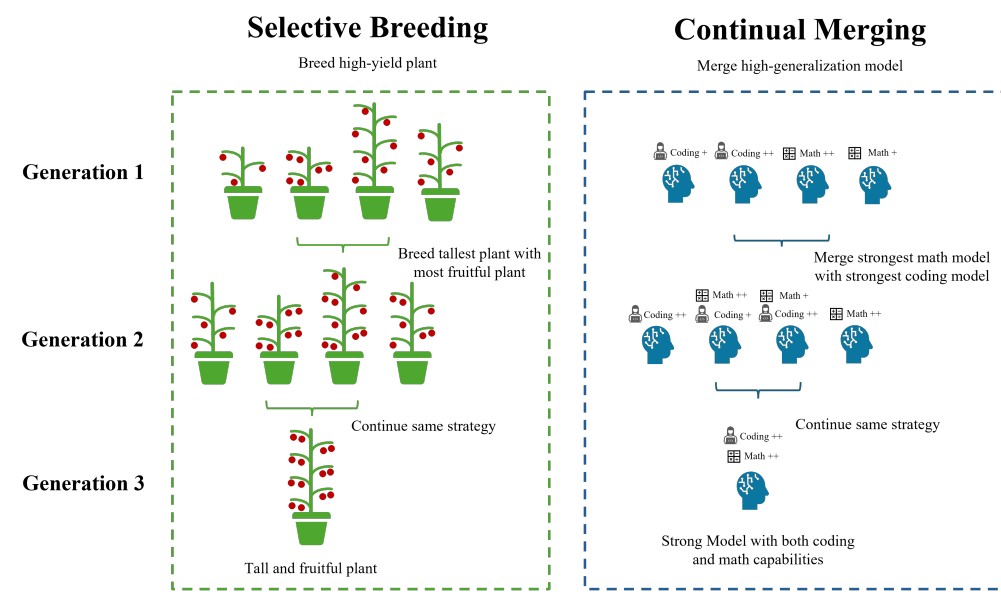

Figure 11: An intuitive **comparison between selective breeding and continual model merging**. The **left** process demonstrates breeding a tall and frutful plant by selecting parents with the desired traits in an biological scenario. The **right** process shows developing a model with capabilities of coding and math through model evolution.

### E.2 INBREEDING DEPRESSION VS. SACUATION STAGE

As highlighted in the main paper, one of our key findings is that the late stage of model evolution often enters a saturation stage, during which models exhibit minimal differences from one another. This phenomenon parallels "inbreeding depression" in artificial selection, where breeding closely related individuals reduces genetic diversity and fitness. Although genetic inheritance and model weights operate differently, merging closely related models leads to new models with minimal variation, thereby reducing the effectiveness of merging, particularly in weight averaging. To address this issue, we propose quantifying the differences between models, a concept we term model kinship, to guide the merging process and mitigate the challenges associated with the saturation stage in model evolution.

## F FULL EVALUATION RESULTS OF MAIN EXPERIMENTS

Table 9 presents detailed evaluation results from the main experiments, while Table 10 provides information on additional experiments conducted using Llama-2. Consistent with the results observed for Mistral-7B, model evolution guided by model kinship produces better generalized models compared to the vanilla greedy strategy in Llama-2.

Table 9: Evaluation Results of Main Experiments of Mistral-7B.

| Model | TruthfulQA | Winogrande | GSM8K | Avg. | Model Kinship |
|---|---|---|---|---|---|
| MetaMath | 44.89 | 75.77 | 70.51 | 63.72 | / |
| Instruct | 68.26 | 77.19 | 40.03 | 61.82 | / |
| Open-chat | 52.15 | 80.74 | 65.96 | 66.28 | / |
| model-1-1-greedy | 52.51 | 76.16 | 57.85 | 62.17 | 0.01 |
| model-1-2-greedy | 58.04 | 76.32 | 57.72 | 64.02 | -0.02 |
| model-1-3-greedy | 48.96 | 78.69 | 72.86 | 66.84 | 0.05 |
| model-2-1-greedy | 50.94 | 80.11 | 75.13 | 68.72 | 0.93 |
| model-2-2-greedy | 49.78 | 78.93 | 55.72 | 61.47 | 0.57 |
| model-2-3-greedy | 52.36 | 78.61 | 52.99 | 61.32 | 0.58 |
| **model-2-exp** | 61.01 | 79.56 | 63.76 | 68.11 | -0.02 |
| model-3-1-greedy | 51.95 | 80.51 | 73.31 | 68.59 | 0.95 |
| model-3-2-greedy | 49.96 | 79.72 | 73.54 | 67.74 | 0.93 |
| model-3-3 | 56.95 | 80.25 | 70.00 | 69.06 | 0.24 |
| model-3-4 | 54.38 | 78.45 | 73.01 | 68.61 | 0.32 |
| **model-3-exp** | 54.13 | 78.69 | 71.65 | 68.15 | 0.03 |
| model-4-1-greedy | 50.82 | 80.11 | 74.60 | 68.51 | 0.98 |
| model-4-2-greedy | 50.36 | 79.47 | 74.31 | 68.04 | 0.98 |
| model-4-3-greedy | 51.04 | 79.72 | 74.83 | 68.53 | 0.94 |
| model-4-4 | 53.31 | 79.40 | 73.54 | 68.75 | 0.54 |
| model-4-5 | 52.48 | 79.01 | 73.68 | 68.39 | 0.66 |
| model-4-6 | 53.69 | 79.72 | 73.69 | 69.03 | 0.52 |
| **model-4-exp** | 55.16 | 78.53 | 71.80 | 68.49 | 0.48 |
| model-5-1 | 54.85 | 79.37 | 73.31 | **69.13** | 0.65 |
| model-5-2 | 54.78 | 79.40 | 72.86 | 68.98 | 0.65 |
| model-5-3 | 53.49 | 79.24 | 73.16 | 68.63 | 0.98 |
| **model-5-exp** | 52.98 | 79.32 | 72.78 | 68.36 | 0.59 |

Table 10: Evaluation Results of addtional experiments of Llama-2.

| Model | TruthfulQA | Winogrande | GSM8K | Avg. | Model Kinship |
|---|---|---|---|---|---|
| winogrande | 42.0 | 77.9 | 6.4 | 42.1 | / |
| GSM8K | 39.0 | 73.4 | 38.0 | 50.1 | / |
| TruthfulQA | 56.7 | 68.9 | 9.5 | 45.0 | / |
| child1-1-greedy | 40.2 | 79.3 | 34.2 | 51.2 | 0.03 |
| child1-2-greedy | 46.7 | 74.4 | 34.2 | 51.7 | 0.01 |
| child1-3-greedy | 46.1 | 77.1 | 1.9 | 41.7 | 0.01 |
| child-2-1-greedy | 44.6 | 78.6 | 36.8 | 53.3 | 0.19 |
| child-2-2-greedy | 43.7 | 74.0 | 40.4 | 52.7 | 0.45 |
| child-2-3-greedy | 38.9 | 77.5 | 37.1 | 51.1 | 0.39 |
| **child-2-exp** | 43.3 | 81.2 | 28.5 | 51.0 | 0.01 |
| child-3-1-greedy | 44.2 | 77.1 | 37.3 | 52.8 | 0.88 |
| child-3-2-greedy | 45.4 | 77.5 | 34.5 | 52.4 | 0.79 |
| child-3-3-greedy | 45.0 | 73.8 | 36.6 | 51.8 | 0.89 |
| **child-3-exp** | 45.1 | 78.6 | 30.3 | 51.3 | 0.58 |
| child-4-1-greedy | 44.4 | 78.5 | 36.8 | 53.2 | 0.95 |
| child-4-2-greedy | 44.1 | 75.5 | 40.0 | 53.1 | 0.97 |
| **child-4-exp** | 43.3 | 80.9 | 32.6 | 52.2 | 0.81 |
| child-5-1-greedy | 44.2 | 77.1 | 37.2 | 52.8 | 0.97 |
| child-5-2-greedy | 44.3 | 77.4 | 36.7 | 52.8 | 0.91 |
| child-5-3-greedy | 44.3 | 78.3 | 36.8 | 53.1 | 0.98 |
| **child-5-exp** | 44.5 | 78.1 | 32.0 | 51.5 | 0.64 |
| child-6-1-greedy | 44.5 | 78.5 | 36.8 | 53.2 | 0.99 |
| child-6-2-greedy | 44.4 | 78.3 | 36.8 | 53.2 | 0.99 |
| child-6-3-greedy | 44.3 | 78.3 | 36.8 | 53.1 | 0.99 |
| **child-6-exp** | 44.3 | 80.4 | 35.3 | **53.4** | 0.80 |

