# OpenReview forum: "Exploring Model Kinship for Merging Large Language Models"
_ICLR.cc/2025/Conference — Submitted to ICLR 2025_

### Official Review · Reviewer_Xidk · 2024-10-22

**Soundness:** 3
**Presentation:** 3
**Contribution:** 3
**Rating:** 6
**Confidence:** 3

**Summary:**

This paper introduces a notion of model kinship, which attempts to measure the relatedness of two language models. The paper also presents an algorithm that leverages the degree of model kinship to inform model merging strategies. In particular, the paper suggests that merging highly performant models with low kinship to a target model outperforms greedily merging with the top performing model when iteratively merging models. The results in the paper suggest that a model kinship aware strategy for model merging leads to superior benchmark results over model kinship unaware strategies.

**Strengths:**

The paper posits and provides evidence for the importance of kinship aware strategies when performing iterative model merging. This is a highly active and impactful area of research, and the paper presents empirical results with sufficient theoretical justification that they should be noted by the broader research community.

**Weaknesses:**

The paper lacks experimental breadth, and in some cases, seems to have critical typos. The paper would benefit from expanded experiments, particularly with additional merge lineage studies and with simulations on base models beyond Mistral 7b.

**Questions:**

086-087 should have a comma after improvements

table 1 and the significance of the p values for absolute gain seem to indicate that the variability of the merge gain is inversely related to the degree of model kinship. it may be worth remarking on this around line 233.

297-299 make references to a “notable correlation” between model kinship and average task performance in figure 4, but no formal correlation analysis is presented.

311 makes reference to 3 learning stages, but line 086 makes reference to only two. this complicates the narrative

345 comma after convergence should be a period.

algorithm 1 has what seems to be some typos:

- should line 2 have M_k, not M_n, as the last element in the set?

- should line 3 replace “top m” with “top k”?

- should line 7 replace “select k pairs” with “select k models?”

- should line 9 be “identify the model M_f \in S with the lowest kinship…”? the discussion on lines 414-416 seems to indicate so.

- line 11 should expand on what “model-gen-id” is

line 429 seems to have an unlinked figure reference (perhaps to figure 6?)

---

> ### Author Response · Authors · 2024-11-20
>
> Thank you very much for your review. We have corrected the typos you highlighted as weaknesses. In the following section, I will address the concerns you raised. If there is other questions, please let us know.
>
> ---
>
> **About notable correlation in line 297-299:**
> Figure 4 and the phrase "notable correlation" are intended to emphasize that model kinship follows a pattern similar to average task performance, reflecting the progression of model evolution. The goal is to introduce a more detailed analysis of model kinship in relation to merging progress, as indicated by average task performance in the next section. Since "notable correlation" conveys the same idea as "follows similar patterns," we have removed it for clarity and conciseness, thus reducing potential confusion.
>
> ---
>
> **About 3 learning stages in line 311:**
> In this section, the three model groups consist of models from two learning stages, consistent with the previous presentation, **along with the initial fine-tuned models** (the foundation models before the merging process). The inclusion of the initial models is intended to provide a more comprehensive view of the entire evolution process. We have revised this section for improved clarity.
>
> ---
>
> **About Algorithm 1:**
> - **Line 2**: It should be **M_k**, indicating the set of k models in the top-k set.
> - **Line 3**: It should be '**top-k**'.
> - **Line 7**: This refers to each model pair from the top-k set, which includes the top-k models but should not include k itself.
> - **Line 9**: We acknowledge that this state is somewhat confusing. The intended meaning is that the current best model is merged with the model having the lowest kinship with the best model from the previous generation. The key concept is to incorporate a model that may not currently perform outstandingly (and could be overlooked by a greedy strategy), but has the lowest kinship with the best model (potentially indicating better performance after merging).
>
> Furthermore, we have updated the notation for enhanced clarity in the revised version of our paper.
>
> ---
>
> **About Figure Reference:**
> Indeed, this refers to Figure 6, and we have made the necessary correction in the revised version.
>
> ---
>
> **About Experimental Breadth:**
> We acknowledge that our experiments are limited in terms of architecture and task sets. We include three additional experiments:
> 1. Evolution experiments on the Llama-2 model
> 2. Ablation experiments for the greedy strategy
> 3. Evolution experiments on alternative task sets
>
> Further details will be provided in the common reply section.

---

### Official Review · Reviewer_fHcA · 2024-10-27

**Soundness:** 3
**Presentation:** 2
**Contribution:** 3
**Rating:** 6
**Confidence:** 2

**Summary:**

1. The paper explores the concept of "model kinship," which measures the similarity or relatedness between large language models (LLMs), drawing a parallel to biological evolution. Through empirical analysis, the authors demonstrate that this kinship can predict the performance gains when merging models, offering a basis for selecting suitable candidates for merging.

2. Building on this, they propose a strategy called Top-k Greedy Merging with Model Kinship, which improves performance on benchmark datasets. The study shows that using model kinship not only helps avoid local optima during model evolution but also serves as an early stopping criterion, enhancing efficiency. The authors suggest that their findings provide valuable insights for future LLM merging and evolution.

**Strengths:**

1. The paper introduces a novel exploration of the kinship between LLMs, using a heuristic approach inspired by biological evolution to guide the selection of models for merging. The overall idea is interesting.

2. The narrative of the paper is well-structured, and the authors validate their experiments using Mistral as the primary architecture.

**Weaknesses:**

1.  The experimental section lacks clarity. It is recommended that the authors add a dedicated subsection analyzing the datasets used and explaining why these specific datasets (e.g., general-purpose or domain-specific test data) were chosen for performance evaluation.

2. In the appendix (Figure 8), the authors present performance analysis based on various merging experiments using the Mistral architecture. However, it is unclear whether the results are applicable to other architectures like LLaMA, which suggests the idea may have some limitations. Although the authors mention in the limitations section that the performance on other architectures remains unknown, I strongly recommend testing on LLaMA or other types of LLM architectures to make the argument more robust and solid.

3. In Algorithm 1, the authors use Greedy Merging with Model Kinship; it is suggested to include an ablation study to demonstrate that the greedy algorithm is indeed the optimal choice.

**Questions:**

1. Please respond to the weaknesses mentioned in my review. I will adjust the score accordingly based on your response.

---

> ### Author Response · Authors · 2024-11-20
>
> Thank you for your thoughtful review. We understand your concerns regarding the generalization of model kinship across datasets and architectures. We are currently conducting additional experiments to validate the generalization of model kinship in various scenarios. Most information will be included in the Common reply section.
>
> ---
>
> **Reason for Selecting the Current Task Set.**
> The datasets used in our experiments are task-specific, chosen to better represent the strengths of foundation models. We selected these three datasets to maintain consistency with the task set employed in **community experiments**, which currently serve as the primary application environment for iterative merging techniques.
>
>
> More specifically, our task set (**{Winogrande, GSM8K, TruthfulQA}**) is a subset of the six key benchmarks ({**ARC, HellaSwag, MMLU, TruthfulQA, Winogrande, GSM8K**}) featured on the Open LLM Leaderboard. These benchmarks were selected for their capacity to assess diverse reasoning abilities and general knowledge across a wide range of fields, making them applicable in various scenarios.
>
> However, we acknowledge that our experiments are limited in task sets. At present, we are working on the addtional experiments including other task sets. Theoretically, as model kinship is rooted in the concept of task vectors, we hypothesize that it is applicable to tasks where task vectors are feasible. Additional evidence will be provided in the common reply section once the ongoing experiments are completed. We will reach out to you for this question at that time.
>
> ---
>
> **Generalization on Llama.**  We provide additional evidence in the Common Reply section regarding the Llama-2 experiment, conducted using the same task set and method as the main experiment. Further details are available in the Common Reply section.
>
> ---
>
> **Ablation study of greedy merging.**
> The motivation behind Greedy Merging is to simulate the intuitive decision-making process of typical users, and it is generally regarded as the baseline strategy. For the ablation study, we implement a random selection approach. Further details are provided in the Common Reply section.

---

> > ### Comment · Reviewer_fHcA · 2024-11-24
> >
> > Thanks, I have raised your score. Good job!

---

> > > ### Author Response · Authors · 2024-11-24
> > > **Thank you!**
> > >
> > > Thank you for reevaluating our work! We are pleased to have addressed your concerns and are committed to improving our research based on your valuable feedback. We will continue to validate its applicability across diverse scenarios to strengthen its contributions.
> > >
> > > We deeply appreciate the time and effort you have dedicated to providing us with such thoughtful suggestions.

---

### Official Review · Reviewer_vep1 · 2024-10-31

**Soundness:** 3
**Presentation:** 3
**Contribution:** 3
**Rating:** 8
**Confidence:** 3

**Summary:**

This paper proposes model kinship to evaluate the degree of similarity between LLMs. Then the authors did experiments to validate that model kinship moderately correlates with merge gain, and found that the performance improvement during model merging can be divided into learning stage and saturation stage. Finally, the paper proposes a practical use case to apply model kinship into top-k greedy merging to improve the performance of model evolution.

**Strengths:**

This paper targets at solving an important task. Also, this paper is very complete and well-written, from introducing the concept of model kinship, to empirically verify its value, and finally provide a practical scenario to apply model kinship to improve the model merging performance. I feel this is a complete work.

**Weaknesses:**

1. The bold fonts in this paper emphasize the relationship between model merging and biological evolution. Actually, I don't quite get their correlation. Biological evolution is based on nature selection while in this paper, model merging is done based on model similarity. Can the authors further elaborate on this?

2. Since all the experiments in this paper are only verified on one model set and one task set, there is no guarantee that the proposed pipeline can be generalized to other real-world scenarios.

**Questions:**

Please see above weakness.

---

> ### Author Response · Authors · 2024-11-20
>
> Thank you very much for your review. We notice that the correlation between biological evolution an model merging have not been fully discussed in the current version of our paper and some presentation is too short to cause confusion. We have added a new section (Appendix.E) in revised version biological concepts comparison.
>
> ---
> **Artificial Selection and Model Evolution.** For biological references, we draw an analogy between **artificial selection** in biological evolution and **model evolution** through model merging. In artificial selection, desired traits are preserved by manually selecting breeding pairs in each generation, typically those exhibiting the most significant features. Similarly, model evolution (iterative model merging) follows a comparable process. Users preserve desired task capabilities by manually selecting merging pairs. Through iterative merging, they can develop a model that excels across all tasks in the desired task set.
>
> **Inbreeding Depression and Saturation Stage.** One of our key findings is that the late stage of model evolution often enters a saturation stage, where models exhibit significantly low differences among themselves. This phenomenon is analogous to "inbreeding depression" in artificial selection, where breeding closely related individuals leads to a reduced gene pool and diminished fitness. Similarly, merging closely related models results in new models with minimal variation, thereby diminishing the benefits of merging, particularly with weight averaging method.
> To address this, we propose measuring the differences between models, termed **model kinship**, to mitigate this issue during model evolution.
>
> ---
>
> **Concern about generalization in task sets**:  We acknowledge that our experiments are limited in terms of architecture and task sets. At present, we are working on the addtional experiments including other task sets. Theoretically, as model kinship is rooted in the concept of task vectors, we hypothesize that it is applicable to tasks where task vectors are feasible. Additional evidence will be provided in the common reply section once the ongoing experiments are completed. We will reach out to you for this question at that time. If there is other questions, please let us know.

---

> > ### Comment · Reviewer_vep1 · 2024-11-25
> >
> > Thanks for the response. After reading the new experimental results, I'm more convinced by the generalization of this proposed method. I don't have any more concerns. Therefore, I raised my score to 8.

---

> > > ### Author Response · Authors · 2024-11-26
> > > **Thank you!**
> > >
> > > Thank you for taking the time to reevaluate our work! We are glad to have addressed your concerns and appreciate your valuable feedback. We are committed to further enhancing our research and will continue to validate its applicability across a variety of scenarios to strengthen its contributions.

---

### Official Review · Reviewer_haon · 2024-11-02

**Soundness:** 3
**Presentation:** 2
**Contribution:** 2
**Rating:** 3
**Confidence:** 3

**Summary:**

This paper introduces the concept of model kinship, drawing an analogy to genetic hybridization in biology, with the aim of providing guidance for effective performance improvements in the field of model merging. The authors first empirically validate the relationship between kinship and performance gains, demonstrating their consistency. Building on the concept of kinship, they propose a Top-k Greedy Merging method that shows some performance enhancements compared to baseline approaches.

**Strengths:**

1. The research area is significant, and the introduction of the kinship metric, drawing an analogy to biological evolution, presents a human-like perspective that adds depth to the discussion of model merging.

2. The paper effectively establishes a connection between kinship and performance gains, providing empirical evidence that supports the proposed approach.

**Weaknesses:**

1. The definition of the problem is unclear, and the paper lacks a thorough discussion of specific scientific issues within the model merging field. The research motivation is insufficient, leading to a lack of logical clarity in the arguments presented.

2. While introducing human-like concepts or analogies from other fields can be a valuable approach to problem-solving, the kinship metric designed in this paper lacks a reasonable formulation and effective validation, which undermines its applicability.

3. The logical coherence and rigor of the writing require significant improvement to enhance the overall quality of the paper.

4. The conclusions drawn in the paper lack persuasive power due to the absence of quantitative experimental analysis. The methodology and experimental design appear relatively crude, which detracts from the credibility of the findings.

**Questions:**

As mentioned in the above weaknesses.

**Details Of Ethics Concerns:**

None.

---

> ### Author Response · Authors · 2024-11-20
>
> Thank you very much for your review. Based on your feedback regarding the first two questions, we have restructured Section 2 (Background) to improve clarity and presentation. New contents are highlighted in the blue color.
> If there is other questions, please let us know.
>
> ---
>
> **Problem Definition and Research Motivation.**
> We focus primarily on the iterative merging process (model evolution). To the best of our knowledge, this technique was first adopted by the Hugging Face merging community with Mistral-7B [1]. The community iteratively refines models through repeated applications of model merging, resulting in the creation of several powerful models, such as **YamShadowExperiment28-7B**.
>
> The table below demonstrates the effectiveness of iterative merging compared to the single-merge method. The results indicate that iterative merging can more effectively acquire generalized models, achieving higher overall performance.
>
> | Model                          | TruthfulQA | Winogrande | GSM8K  | Average |
> |--------------------------------|------------|------------|--------|---------|
> | MetaMath-Mistral-7B            | 44.89      | 75.77      | 70.51  | 63.72   |
> | Mistral-7B-Instruct-v0.2       | 68.26      | 77.19      | 40.03  | 61.82   |
> | Open-Chat-3.5-1210             | 52.15      | 80.74      | 65.96  | 66.28   |
>
> | Method                         | TruthfulQA | Winogrande | GSM8K  | Average |
> |--------------------------------|------------|------------|--------|---------|
> | Linear weight averaging (1:1:1)| 56.37      | 78.08      | 67.54  | 67.33   |
> | Model-3-3 (Model Evolution)    | 56.95      | 80.25      | 70.00  | 69.06   |
>
> However, iterative merging lacks a general merging strategy, which limits its overall effectiveness. Common methods often employ greedy strategies, such as merging models with the highest average performance. While this approach is relatively effective, our observations indicate that the community frequently attempts to merge models with nearly identical weights. This practice leads to resource waste and stagnation, as no further improvements are achieved. Our work aims to develop a more effective metric to guide iterative merging, potentially avoiding self-merging and enabling further advancements.
>
> ---
>
> **Reason of Kinship Metric Formulation.**
> Model kinship leverages the structure of task vectors [2], which have proven effective in representing task specific information. Our approach extends this idea to a more complex multi-task scenario. Here, the similarity between the delta parameters (task vectors) of two models, calculated using similarity metrics (three of which have been tested in our analysis), can reveal differences in overall task capabilities, thereby enhancing the quality of the merging process.
>
> In the revised version of our paper, we have incorporated additional background to clarify the motivation for formulating model kinship.
>
> ---
>
> **Quantitative Analysis to validate Model Kinship.**
> We agree that more quantitative analyses are necessary to validate the concept of model kinship and support our conclusions. Since iterative merging is a novel concept in the field of model merging, existing experiments are limited. To address this, we conducted additional experiments to evaluate the generalization of model kinship across architectures and task sets.
>
> Details of these experiments are provided in the common section. Note that the experiments are still ongoing, and we will provide updates upon its completion.
>
> ---
>
> **References**
>
> [1] Huggingface Open LLM Leaderboard - v1
>
> [2]
> Gabriel Ilharco et al. Editing Models with Task Arithmetic. ICLR 2023

---

> > ### Author Response · Authors · 2024-12-02
> > **Kindly Remind**
> >
> > Thank you for your time and valuable feedback on our work.  **As the reviewer reply period is coming to the end**, we kindly encourage you to share any remaining questions or concerns at your earliest convenience, as we will not be able to respond after this period.
> >
> > Your insights are greatly appreciated and have significantly contributed to improving our paper.

---

> ### Comment · Reviewer_haon · 2024-11-25
>
> Thank you for your effort in revising and enhancing the manuscript. I truly appreciate the improvements made. However, after a closer examination, I still have several questions and concerns that I hope the authors can address to clarify and strengthen the arguments presented in the paper:
>
> 1. On the relationship between Kinship and performance gain:
>
> (1) Support for the qualitative conclusion in Section 3.2:
> In Section 3.2, the manuscript suggests a positive relationship between Kinship and Performance Gain. However, the experiments do not seem to explicitly validate this assertion. While the Kinship concept is introduced to “identify task-related differences between models to maximize the outcomes of merging,” the conclusions drawn in Section 3.2 appear insufficient to substantiate this claim. Could the authors provide additional experimental evidence or a clearer explanation to reinforce this point?
>
> (2) Theoretical basis of Kinship in Section 2.3:
> The definition of Kinship as the similarity of model weight differences is stated to reflect “the similarity of task capabilities between models.” However, it is unclear why weight difference similarity necessarily correlates with task capability similarity. Could the authors elaborate on the theoretical foundation for this claim? Furthermore, are there potential counterexamples (e.g., models with similar weights but divergent task performance) that could challenge this assertion?
>
> (3) Key statement in Section 3.2.1:
> The manuscript suggests that Kinship “may serve as a key factor in determining the upper limit of merge gains, thus indicating potential improvements.” While this statement is intriguing, it lacks direct experimental validation. Could the authors provide quantitative evidence demonstrating that Kinship reliably predicts the upper limit of performance gains, perhaps through controlled experiments or ablation studies?
>
> 2. On the main experimental results:
>
> (1) Criteria for stage division in Section 3.4:
> In Section 3.4, the thresholds for “Saturation Stage (≥ 0.75)” and “Learning Stage (< 0.75 and ≥ 0.73)” are introduced. However, the basis for this division is not clearly explained. Are these thresholds derived from statistical analysis, theoretical insights, or empirical observations? If they are empirically determined, additional experimental justification would be helpful. If they are theoretically motivated, the rationale should be explicitly stated.
>
> (2) Relationship between Kinship and higher average task performance:
> The manuscript claims that “model Kinship increases with the average task performance, even across models that follow different evolution paths yet share similar average performance levels.” However, this relationship is not sufficiently clarified. For instance, the proximity of parameter similarity near a local optimum (using the base model as the initialization path) might merely reflect optimization path adjacency rather than providing evidence for the core conclusion. Could the authors provide a more rigorous explanation or analysis to strengthen this argument?
>
> Summary and Suggestions:
> To enhance the clarity and rigor of the manuscript, I suggest the following:
>
> Provide stronger theoretical and experimental support for the definition of Kinship and its contribution to performance improvements.
> Elaborate on the causal relationships and logical connections in the main experimental results, especially regarding the relationship between Kinship and performance gains.
> Offer more detailed explanations or acknowledge potential limitations in the current analysis to improve the overall scientific rigor.

---

> > ### Author Response · Authors · 2024-11-29
> >
> > We would like to express our sincere gratitude to your suggestions of improvements.
> > To address your concerns, we have revisited the preliminary analysis reported in **Section 3**, refining confusing statements and improving logical connections.
> > In the following sections, we will discuss the observations derived from the empirical evidence in **Section 3**.
> >
> >
> > ---
> >
> > ## The Aims of Section 3
> >
> > Section 3 aims to provide motivation for the application of model kinship (introduced as an exploration step in the main experiments in **Section 4**) based on empirical evidence from community experiments. Specifically, we identify two key observations:
> >
> > - **Section 3.2** - The relationship between model kinship and merge gain: This suggests a potential approach to leveraging model kinship.
> > - **Section 3.3, 3.4** - The relationship between model kinship and average task performance (as an indicator of merging evolution stages): This highlights a possible challenge in common model evolution.
> >
> > Our main focus is to demonstrate the application of model kinship. As such, we do not delve deeper into validating these two observations in the current work. However, we acknowledge that thoroughly investigating these observations could significantly enhance our understanding of model merging, strengthen our main method, and inspire new directions for future research.

---

> > ### Author Response · Authors · 2024-11-29
> >
> > ## Question 1: Model Kinship and Performance Gain
> > The aim of model kinship—**"to identify task-related differences between models to maximize the outcomes of merging"** involve two primary requirements:
> >
> > 1. **Ability to represent task differences.**
> > 2. **Ability to enhance the effectiveness of merging.**
> >
> > The examination of these two requirements in the community experiments data is presented in **Section 3**, focusing on the current model kinship.
> > However, the precise application and validation are provided through our main experimental results in **Section 4**
> >
> > ### **Q1.1,Q1.3**: Upper limit observation and Leveraging model kinship to maximize the outcomes of merging
> > The relationship between model kinship and merge gain is first observed in Figure 2.
> > The correlation is observed through statistical analysis, while the scatter plots clearly demonstrate an upper boundary, suggesting a potential relationship between model kinship and the upper limit of merge gain
> > Based on these findings, we hypothesize that lower kinship in merges may result in greater merge gains, potentially addressing the challenge of the greedy strategy (as discussed in Sections 3.3 and 3.4).
> > Therefore, in the main experiment, we design the 'exploration by model kinship' steps, and the main results show that it can help achieve superior final models, corresponding to **"maximize the outcomes of merging"**.
> >
> > Additionally, similar observation of correlation between model kinship and performance gain can be found in our experimental results.
> >
> > | Model Name  | Merge Gain | Model Kinship |
> > |-------------|------------|---------------|
> > | model-1-2   | -0.03      | -0.02         |
> > | model-2-exp | 3.78       | -0.02         |
> > | model-1-1   | -0.6       | 0.01          |
> > | model-3-exp | 2.7        | 0.03          |
> > | model-1-3   | 1.84       | 0.05          |
> > | model-3-3   | 0.74       | 0.24          |
> > | model-3-4   | 1.13       | 0.32          |
> > | model-4-exp | 3.42       | 0.48          |
> > | model-4-6   | 0.15       | 0.52          |
> > | model-5-exp | -0.09      | 0.53          |
> > | model-4-4   | -0.14      | 0.54          |
> > | model-2-2   | -3.96      | 0.57          |
> > | model-2-3   | -3.83      | 0.58          |
> > | model-5-1   | 0.04       | 0.65          |
> > | model-5-2   | 0.07       | 0.65          |
> > | model-4-5   | -0.27      | 0.66          |
> > | model-2-1   | 2.16       | 0.93          |
> > | model-3-2   | 1.09       | 0.93          |
> > | model-4-3   | 0.37       | 0.94          |
> > | model-3-1   | 1.09       | 0.95          |
> > | model-4-1   | -0.14      | 0.98          |
> > | model-4-2   | -0.19      | 0.98          |
> > | model-5-3   | -0.37      | 0.98          |
> >
> > To explain the rationale behind their connections, we focus on the selection of three model merges: model4-1(-0.14, 0.98), model2-exp(3.78, -0.02), and model2-1(2.16, 0.93). Each of these models exhibits distinct characteristics in terms of both model kinship and task performance.
> >
> > | Model Name  | TruthfulQA | Winogrande | GSM8K |
> > |-------------|------------|------------|-------|
> > | model2-1    | 50.94      | 80.11      | 75.13 |
> > | model3-2    | 49.96      | 79.72      | 73.54 |
> > | **model4-1**    | 50.82      | 80.11      | 74.6  |
> >
> > | Model Name  | TruthfulQA | Winogrande | GSM8K |
> > |-------------|------------|------------|-------|
> > | Instruct    | 68.26      | 77.19      | 40.03 |
> > | model1-3    | 48.96      | 78.69      | 72.86 |
> > | **model2-exp**  | 61.01      | 79.56      | 63.76 |
> >
> > | Model Name  | TruthfulQA | Winogrande | GSM8K |
> > |-------------|------------|------------|-------|
> > | Open-chat   | 52.15      | 80.74      | 65.96 |
> > | model1-3    | 48.96      | 78.69      | 72.86 |
> > | **model2-1**    | 50.94      | 80.11      | 75.13 |
> >
> >
> > - **Model4-1** represents a high kinship merge, where task performance across models is very close. This lack of significant differentiation limits the merge's ability to effectively learn from each model, resulting in a low merge gain.
> >
> > - **Model2-exp** demonstrates a low kinship merge, where there is a noticeable difference in task performance between the models, leading to a higher merge gain.
> >
> > These two instances illustrate the most common scenarios of how **model kinship** affects **merge gain**.
> >
> > However, we also observe **Model2-1**, which demonstrates a **high kinship merge** but achieves a certain degree of **merge gain**. In this case, the merging process prioritizes the model with **superior task capabilities**. Unlike **Model2-exp**, however, it does not result in a task capability that surpasses that of any parent model (e.g., in **Winogrande**).
> >
> > We hypothesize that this behavior may be connected to the **"weak-to-strong" method** (referenced in [2]).
> > Rigorously investigating this relationship would be highly beneficial for future study.
> > However, conducting such comprehensive experiments within the discussion period is not feasible, as it would require further tuning and testing across diverse task set scenarios.

---

> > ### Author Response · Authors · 2024-11-29
> >
> > ### Q1.2: Model kinship and Task Relationship
> >
> > In the formulation of model kinship, we use the placeholder $ sim(⋅,⋅) $ as a similarity metric function to explore options that can effectively capture task-related differences.
> > One such metric is cosine similarity, derived from the analysis in task vector paper [1], which has been validated as effective for representing differences in single-task models through the cosine similarity of delta parameters (task vectors). In addition to cosine similarity, we also investigate the Pearson correlation coefficient and Euclidean distance.
> >
> > However, we acknowledge that we have not thoroughly evaluated the applicability of these metrics in the context of model evolution, particularly for merged models with multitask capabilities.
> > To address this, we examine the relationship between the similarity metrics and task information in subsequent sections.
> >
> > Our analysis focuses on the LLaMA-2 architecture, as we can find the necessary open source fine-tuned checkpoints on various datasets.
> > To measure differences between models, we currently use a preliminary evaluation method: the **Average Task Performance Difference** (ATPD), which aims to represent task capability differences based on evaluation performance.
> >
> > The Average Task Performance Difference (ATPD) between two models, $ M_1 $ and $ M_2 $, is calculated by averaging the absolute differences in performance across all tasks. Let $ T $ denote the set of tasks, and $ P_i^{(j)} $ represent the performance of model $ M_j $ on task $ i $.
> > Then, the ATPD is defined as:
> >
> > $$
> > \text{ATPD}(M_1, M_2) = \frac{1}{|T|} \sum_{i \in T} \left| P_i^{(1)} - P_i^{(2)} \right|
> > $$
> >
> > - $ |T| $: the total number of tasks.
> > - $ P_i^{(1)} $ and $ P_i^{(2)} $: performances of models $ M_1 $ and $ M_2 $ on task $ i $.
> > - $ \left| P_i^{(1)} - P_i^{(2)} \right| $: absolute difference in performance for task $ i $.

---

> > ### Author Response · Authors · 2024-11-29
> >
> > For this study, we utilize models from additional **LLaMA-2** experiments. These models were merged from three fine-tuned models, allowing us to control the generated models to focus solely on the corresponding task capabilities.
> > The following table presents the results, with Winogrande, TruthfulQA, and GSM8K representing the performance differences across each task.
> >
> >
> > | Model 1          | Model 2          | Winogrande | Truthfulqa | GSM8K | ATPD  | Kinship(cs) | Kinship(pcc) | Kinship(ed) |
> > |------------------|------------------|------|------|------|-------|-------------|--------------|-------------|
> > | child-4-1-greedy | child-5-3-greedy | 0.10 | 0.00 | 0.20 | 0.10  | 0.99        | 0.99         | 2.17        |
> > | child-2-1-greedy | child-4-1-greedy | 0.20 | 0.10 | 0.00 | 0.10  | 0.98        | 0.99         | 4.22        |
> > | child-2-1-greedy | child-5-3-greedy | 0.10 | 0.10 | 0.20 | 0.13  | 0.99        | 0.99         | 2.19        |
> > | child-4-exp      | child-2-1-greedy | 1.10 | 0.90 | 0.10 | 0.70  | 0.80        | 0.75         | 25.53       |
> > | child-2-1-greedy | child-3-1-greedy | 0.20 | 1.30 | 0.70 | 0.73  | 0.95        | 0.98         | 6.74        |
> > | child-4-1-greedy | child-6-exp      | 0.10 | 1.90 | 1.40 | 1.13  | 0.74        | 0.71         | 25.54       |
> > | child-4-1-greedy | child-4-2-greedy | 0.30 | 3.00 | 3.20 | 2.17  | 0.97        | 0.98         | 6.57        |
> > | child-2-2-greedy | child-3-1-greedy | 0.50 | 3.10 | 3.10 | 2.23  | 0.97        | 0.98         | 6.57        |
> > | child-2-1-greedy | child-4-2-greedy | 0.50 | 3.10 | 3.20 | 2.27  | 0.91        | 0.96         | 9.29        |
> > | child-3-exp      | child-2-1-greedy | 0.70 | 0.20 | 6.30 | 2.40  | 0.64        | 0.52         | 35.52       |
> > | child-4-exp      | child-2-1-greedy | 1.10 | 2.50 | 4.00 | 2.53  | 0.78        | 0.75         | 25.53       |
> > | child-2-1-greedy | child1-2-greedy  | 2.30 | 4.00 | 2.40 | 2.90  | 0.79        | 0.89         | 15.75       |
> > | child-2-1-greedy | child-2-2-greedy | 0.70 | 4.40 | 3.80 | 2.97  | 0.88        | 0.95         | 12.43       |
> > | child-2-2-greedy | child1-2-greedy  | 3.00 | 0.40 | 6.20 | 3.20  | 0.89        | 0.92         | 11.68       |
> > | child1-1-greedy  | GSM8K            | 1.20 | 5.90 | 3.80 | 3.63  | 0.39        | 0.46         | 36.39       |
> > | child1-1-greedy  | child1-2-greedy  | 6.50 | 4.90 | 0.00 | 3.80  | 0.19        | 0.16         | 38.07       |
> > | child-2-exp      | child-2-1-greedy | 1.10 | 2.80 | 8.10 | 4.00  | 0.58        | 0.77         | 28.33       |
> > | child1-2-greedy  | GSM8K            | 7.70 | 1.00 | 3.80 | 4.17  | 0.45        | 0.38         | 26.32       |
> > | child-2-1-greedy | child1-3-greedy  | 7.80 | 3.10 | 2.90 | 4.60  | 0.58        | 0.51         | 45.24       |
> > | child-3-1-greedy | child-2-exp      | 0.90 | 4.10 | 8.80 | 4.60  | 0.58        | 0.63         | 32.45       |
> > | winogrande       | TruthfulQA       | 14.70 | 9.00 | 3.10 | 8.93  | 0.01        | 0.01         | 74.49       |
> > | child1-2-greedy  | child1-3-greedy  | 0.60 | 2.70 | 32.30 | 11.87 | 0.64        | 0.52         | 46.06       |
> > | child1-2-greedy  | winogrande       | 4.70 | 3.50 | 27.80 | 12.00 | 0.01        | 0.02         | 55.89       |
> > | winogrande       | GSM8K            | 3.00 | 4.50 | 31.60 | 13.03 | 0.03        | 0.11         | 54.01       |
> > | child1-1-greedy  | child1-3-greedy  | 5.90 | 2.20 | 32.30 | 13.47 | 0.52        | 0.64         | 44.16       |
> > | GSM8K            | TruthfulQA       | 17.70 | 4.50 | 28.50 | 16.90 | 0.01        | 0.01         | 61.56       |
> >
> > For comparison, we compute the Pearson Correlation Coefficient between ATPD and each model kinship.
> >
> > | Correlation(cs) | Correlation(pcc) | Correlation(ed) |
> > |-----------------|------------------|-----------------|
> > | -0.77           | -0.74            | 0.80            |
> >
> >
> > The results demonstrate strong correlations: Cosine Similarity (-0.77) and Pearson Correlation Coefficient (-0.74) exhibit negative correlations, while Euclidean Distance (0.80) shows a positive correlation.
> > This support that model kinship is related to task differences.
> > As mentioned in the limitations, the current metrics are viable but not optimal. Combining them with task information studies [2] could further enhance the value of our work.

---

> > ### Author Response · Authors · 2024-11-29
> >
> > ## Question2: Model Kinship and Average Task Performance
> >
> > ### Q2.1 Criteria for stage division
> >
> > **The criteria for stage division** are based on statistical analysis rather than strict thresholds.
> > We select the change points identified in the Section 3.3 results.
> > We aim to use minimal matrices in report to capture the key differences in model kinship among candidate models for merging during the initial merging phase and the two distinct stages, emphasizing the potential challenges in optimization.
> > However, for robust validation, introducing a more concrete division and increasing the scale of the matrix would be beneficial.
> > We will further investigate this in future studies.
> >
> > ### Q2.2 Relationship between Kinship and higher average task performance
> > In **Section 3.3**, we observe that along an evolution path, both model kinship and average task performance increase as the merging process progresses.
> > Additionally, we find that model kinship exhibits a similar two-stage pattern, particularly prominent in Path 1.
> > **Section 3.4** seeks to further investigate and generalize the observations of model kinship and the stages described in **Section 3.3**.
> > Specifically, it examines whether model kinship is related to the saturation problem.
> > Our results show that the model kinship matrix indicates an increasing kinship between random candidate models throughout evolution.
> > Notably, after **MonaTriX-v6**, we observe a high mutual model kinship exceeding 0.95.
> >
> > It is important to note that **optimization path adjacency** is indeed the explanation we adopt.
> > Based on the mathematical nature of model kinship, which reflects weight-space relatedness, we assume that the greedy strategy may cause optimization path to diverge.
> >
> > Therefore, we approach the main experiments (**Section 4**) from an optimization perspective, hypothesizing that greedy merging may lead to local optima in model evolution.
> > To address this, we introduce an exploration step designed to avoid optimization path adjacency and expand the search space for potentially better solutions.
> > The main experiments successfully reproduce the saturation, and our exploration step identifies better generalized models, supporting our hypothesis.
> > Building on this, we propose in **Appendix C** a hypothesis regarding the optimization process to explain the mechanism underlying model evolution.

---

> > ### Author Response · Authors · 2024-11-29
> >
> > **References**
> >
> > [1]
> > Gabriel Ilharco et al. Editing Models with Task Arithmetic. ICLR 2023
> >
> > [2]
> > Localizing Task Information for Improved Model Merging and Compression. ICML 2024
> >
> > [3]
> > Weak-to-Strong Extrapolation Expedites Alignment. CoRR 2024

---

### Author Response · Authors · 2024-11-20
**Global Response**

We sincerely thank all reviewers for their valuable feedback on our paper.

---

# Our strength

We appreciate the reviewers for recognizing our strengths.

### **Comprehensive Work**

The paper presents a complete and well-written study, effectively introducing the concept of model kinship, empirically verifying its value, and demonstrating its practical application to enhance model merging performance. (Reviewer vep1)

### **Novel and Heuristic Approach**


The introduction of the kinship metric, inspired by biological evolution, enriches the discussion of model merging with a human-like perspective. (Reviewer haon)

The exploration of kinship between LLMs using a heuristic inspired by biological evolution is innovative. (Reviewer fHcA)

### **Impactful and Empirical Evidence**

The paper highlights the significance of kinship-aware strategies in iterative model merging, supported by empirical evidence and theoretical insights in a highly active and impactful research area. (Reviewer Xidk)

The paper establishes a clear link between model kinship and performance gains, supported by empirical evidence for the proposed approach. (Reviewer haon)

---

# Our revisions and additional experiments

We also thank the reviewers for highlighting potential issues. In response to their concerns and questions, we have revised our paper, with modifications highlighted in blue. These updates include:

- Enhance the problem definition and refine the concept formulation in the background section. (Reviewer Haon) (Page 2-3)
- Provide a detailed explanation of the datasets used in the experimental settings. (Reviewer fHcA) (Page 7)
- Improve the presentation of the algorithms employed in the main experiments. (Reviewer Xidk) (Page 8)
- Include Appendix E to explain biology-related concepts. (Reviewer Vep1) (Page 21)
- Correct the typos identified in the previous version. (Reviewer Xidk)

Notably, concerns regarding the **generalization of our method** are raised by most reviewers. To address these concerns, we present additional experiments in this section. However, due to discussion time constraints, we may not be able to cover all experiments comprehensively.

**Addtional experiments** are conducted in the following order:
- 1.Evolution experiments on the Llama-2 model
- 2.Ablation study for the greedy strategy
- 3.Evolution experiments on alternative task sets

---

> ### Author Response · Authors · 2024-11-20
> **Evolution experiment on Llama-2 model (part1)**
>
> We conduct our experiments on the Llama-2 architecture, adhering to the same settings outlined in the main experiments of our paper.
> The foundation models used are fine-tuned on the **TruthfulQA, Winogrande, and GSM8K** datasets.
>
> The following table presents the evaluation results. Each column represents:
>
> - **Model:** The name of each model. Note that the first three entries are fine-tuned foundation models used in our experiments.
> - **TruthfulQA_mc2, Winogrande, GSM8K:** The benchmark results for each dataset, indicating the model's task-specific capabilities.
> - **Average:** The average score across all benchmarks, reflecting the model's overall generalization performance.
> - **Model Kinship:** The kinship score (Here, we use cosine similarity to measure model kinship) of the parent models involved in the merge, indicating their relatedness.
> - **Parent-1 and Parent-2:** The names of the parent models used in the merging process.
>
> Additionally, '-greedy' indicates that the model is generated by the greedy algorithm, while '-exp' indicates that the model is generated during the exploration step.
>
>
> | Model            | TruthfulQA_mc2 | Winogrande | GSM8K | Average | Model Kinship | Parent-1 | Parent-2       |
> |------------------|----------------|------------|-------|--------|------------|----------|----------------|
> | winogrande*      | 42.0           | 77.9       | 6.4   | 42.1   | /          | /        | /              |:
> | GSM8K*           | 39.0           | 73.4       | 38.0  | 50.1   | /          | /        | /              |
> | TruthfulQA*      | 56.7           | 68.9       | 9.5   | 45.0   | /          | /        | /              |
> | child1-1-greedy  | 40.2           | 79.3       | 34.2  | 51.2   | 0.03       | winogrande | gsm8k          |
> | child1-2-greedy  | 46.7           | 74.4       | 34.2  | 51.7   | 0.01       | TruthfulQA | gsm8k          |
> | child1-3-greedy  | 46.1           | 77.1       | 1.9   | 41.7   | 0.01       | winogrande | TruthfulQA     |
> | child-2-1-greedy | 44.4           | 78.4       | 36.6  | **53.1**   | 0.19       | child1-2-greedy | child1-1-greedy              |
> | child-2-2-greedy | 43.7           | 74.0       | 40.4  | 52.7   | 0.45       | child1-2-greedy | gsm8k          |
> | child-2-3-greedy | 38.9           | 77.5       | 37.1  | 51.1   | 0.39       | child1-1-greedy | gsm8k          |
> | child-2-exp      | 43.3           | 81.2       | 28.5  | 51.0   | 0.01       | child1-2-greedy | winogrande     |
> | child-3-1-greedy | 44.2           | 77.1       | 37.3  | 52.8   | 0.88       | child2-1-greedy | child2-2-greedy              |
> | child-3-2-greedy | 45.4           | 77.5       | 34.5  | 52.4   | 0.79       | child2-1-greedy | child1-2-greedy              |
> | child-3-3-greedy | 45.0           | 73.8       | 36.6  | 51.8   | 0.89       | child2-2-greedy | child1-2-greedy              |
> | child-3-exp      | 45.1           | 78.6       | 30.3  | 51.3   | 0.58       | child2-1-greedy | child1-3-greedy              |
> | child-4-1-greedy | 44.2           | 78.3       | 36.6  | 53.0   | 0.95       | child2-1-greedy   | child3-1-greedy              |
> | child-4-2-greedy | 43.9           | 75.3       | 39.8  | 52.9   | 0.97       | child2-2-greedy   | child3-1-greedy              |
> | child-4-exp      | 43.3           | 80.9       | 32.6  | 52.2   | 0.81       | child2-exp | child2-1-greedy              |
> | child-5-1-greedy | 44.2           | 77.1       | 37.2  | 52.8   | 0.97       | child4-1-greedy   | child4-2-greedy          |
> | child-5-2-greedy | 44.3           | 77.4       | 36.7  | 52.8   | 0.91       | child2-1-greedy   | child4-2-greedy              |
> | child-5-3-greedy | 44.3           | 78.3       | 36.8  | 53.1   | 0.98       | child2-1-greedy   | child4-1-greedy              |
> | child-5-exp      | 44.5           | 78.1       | 32.0  | 51.5   | 0.64       | child3-exp | child2-1-greedy              |
> | child-6-1-greedy | 44.3           | 78.3       | 36.6  | 53.0   | 0.99       | child4-1-greedy   | child2-1-greedy              |
> | child-6-2-greedy | 44.2           | 78.1       | 36.6  | 53.0   | 0.99       | child4-1-greedy   | child5-3-greedy              |
> | child-6-3-greedy | 44.1           | 78.1       | 36.6  | 52.9   | 0.99       | child5-3-greedy   | child2-1-greedy              |
> | child-6-exp      | 44.3           | 80.2       | 35.2  | **53.3**   | 0.80       | child4-exp | child2-1-greedy              |
>
> In the Llama-2 experiments, the kinship-based method outperforms the vanilla greedy method (**e.g., Child-6-Exp > Child-2-1-Greedy**). The greedy method becomes trapped in a local optimum by Generation-6, whereas the kinship-based exploration identifies a new model with better overall performance.

---

> ### Author Response · Authors · 2024-11-20
> **Evolution experiment on Llama-2 model (part2)**
>
> Notably, the related exploration models (Exp-2 in both Mistral-7b and Llama-2) exhibit significantly different strengths across task sets compared to other models within the same generation. We hypothesize that this distinction is a key factor contributing to the effectiveness of model kinship-based exploration.
>
> | **Model**       | **ID**               | **TruthfulQA_mc2** | **Winogrande** | **GSM8K** |
> |-------------|------------------|----------------|------------|-------|
> | Mistral-7B  | child-2-exp      | **61.0**       | 79.5       | 63.7  |
> | Mistral-7B  | child-2-1-greedy | 50.9           | 80.1       | 75.1  |
> | Mistral-7B  | child-2-2-greedy | 49.7           | 78.9       | 55.7  |
> | Mistral-7B  | child-2-3-greedy | 52.3           | 78.6       | 52.9  |
>
> | **Model**       | **ID**               | **TruthfulQA_mc2** | **Winogrande** | **GSM8K** |
> |-------------|------------------|----------------|------------|-------|
> | Llama-2     | child-2-exp      | 43.3           | **81.2**   | 28.5  |
> | Llama-2     | child-2-1-greedy | 44.4           | 78.4       | 36.6  |
> | Llama-2     | child-2-2-greedy | 43.7           | 74.0       | 40.4  |
> | Llama-2     | child-2-3-greedy | 38.9           | 77.5       | 37.1  |

---

> ### Author Response · Authors · 2024-11-22
> **Ablation study for the greedy strategy**
>
> We conduct our ablation experiment on the Mistral-7B architecture, adhering to the same settings outlined in the main experiments of our paper.
>
> The only difference is that we use the **random-merge strategy**, where models in each generation are merged with random models (excluding themselves) from the repository (as illustrated in our main figure).
>
> The following table presents the evaluation results. Each column represents:
>
> - **Model:** The name of each model. Note that the first three entries are fine-tuned foundation models used in our experiments.
> - **TruthfulQA_mc2, Winogrande, GSM8K:** The benchmark results for each dataset, indicating the model's task-specific capabilities.
> - **Average:** The average score across all benchmarks, reflecting the model's overall generalization performance.
> - **Model Kinship:** The kinship score (Here, we use cosine similarity to measure model kinship) of the parent models involved in the merge, indicating their relatedness.
> - **Parent-1 and Parent-2:** The names of the parent models used in the merging process.
>
> | Model                  | TruthfulQA_mc2 | Winogrande | GSM8K  | Average   | Model Kinship | Parent-1  | Parent-2  |
> |------------------------|----------------|------------|--------|-----------|---------------|-----------|-----------|
> | MetaMath-mistral-7B    | 44.89          | 75.77      | 70.51  | 63.72     | /             | /         | /         |
> | Mistral-7B-Instruct-v0.2 | 68.26         | 77.19      | 40.03  | 61.82     | /             | /         | /         |
> | Open-chat-3.5-1210     | 52.15          | 80.74      | 65.96  | 66.28     | /             | /         | /         |
> | child1-1               | 52.51          | 76.16      | 57.85  | 62.17     | 0.01          | Instruct  | MetaMath  |
> | child1-2               | 58.04          | 76.32      | 57.72  | 64.02     | 0.01          | Instruct  | Openchat  |
> | child1-3               | 48.96          | 78.69      | 72.86  | 66.84     | 0.03          | Openchat  | MetaMath  |
> | child2-1               | 44.68          | 74.0       | 50.8   | 56.4      | 0.29          | child1-1  | MetaMath  |
> | child2-2               | 49.78          | 78.93      | 55.72  | 61.47     | 0.41          | child1-2  | child1-3  |
> | child2-3               | 61.01          | 79.56      | 63.76  | 68.11     | 0.01          | child1-3  | Instruct  |
> | child3-1               | 51.52          | 78.23      | 56.71  | 62.15     | 0.84          | child2-1  | child1-2  |
> | child3-2               | 43.52          | 75.22      | 47.43  | 55.39     | 0.59          | child2-2  | MetaMath  |
> | child3-3               | 54.32          | 78.53      | 72.81  | **68.55** | 0.28          | child2-3  | child1-3  |
> | child4-1               | 55.32          | 78.41      | 56.23  | 63.32     | 0.54          | child3-1  | child2-3  |
> | child4-2               | 50.53          | 78.42      | 57.65  | 62.20     | 0.86          | child3-2  | child1-2  |
> | child4-3               | 53.45          | 79.31      | 72.65  | 68.47     | 0.67            | child3-3  | Openchat  |
>
> In the **random-merge strategy**, the average performance in each generation fluctuates. The highest average performance achieved is 68.55, which is lower than the 68.72 obtained in the **greedy experiment**.
> Although the random-merge strategy avoids falling into local optima, its unstable search process may make it inefficient.

---

### Meta-Review · Area_Chair_BCeP · 2024-12-19

**Metareview:**

The paper introduces a notion of model kinship, inspired by concepts in biological evolution, measuring how two language models are related. The authors draw a connection between this metric and the gain from merging model, and propose an algorithm exploiting this connection. The reviews agree that the solved problem is important and appreciate the novelty of the proposed metric. The main concern raised in the reviews relates to the experiments. The reviewers mentioned (1) a lack of clarity in the experiment presentation, (2) that the existing experiments are too limited in scope, and (3) the experiments should better prove the claims made in the paper.

The authors provided a thorough rebuttal, clarifying the vague issues in the existing experiments and adding new experiments giving a better proof of generalizability of their claims. While some of the reviewers were convinced by this response and raised their score, others are still concerned that the empirical section requires more work.

Given the remaining mixed reviews, and the large scope of the changes required, I think its best to be cautious and have the paper go through a new round of reviews. The paper has potential, especially with the added content, but the change it requires seems to large to be done without being properly reviewed.

**Additional Comments On Reviewer Discussion:**

Most reviewers were convinced by the rebuttal, though remained with a low confidence (2 or 3) or weak score (two weak accepts, one accept). One reviewer (haon) was not satisfied with the provided experiments and claims there is a need for a deeper analysis despite the additional experiments. To me, the combination of low confidence, large revision required and mixed scores make this too risky, and it could be better for the paper to go through a new round of reviews

---

### Decision · Program_Chairs · 2025-01-22

Reject